# ARL5b inhibits human rhinovirus 16 propagation and impairs macrophage-mediated bacterial clearance

Suzanne Faure-Dupuy [1,10], Jamil Jubrail [1,2,10], Manon Depierre[1,11], Kshanti Africano-Gomez[1,11], Lisa Öberg [3], Elisabeth Israelsson[3], Kristofer Thörn[3], Cédric Delevoye [4,5], Flavia Castellano [1,6], Floriane Herit [1], Thomas Guilbert[1], David G Russell[7], Gaell Mayer[8], Danen M Cunoosamy [9], Nisha Kurian [9] & Florence Niedergang [1✉]

## Abstract

**Human rhinovirus is the most frequently isolated virus during severe exacerbations of chronic respiratory diseases, like chronic obstructive pulmonary disease. In this disease, alveolar macrophages display significantly diminished phagocytic functions that could be associated with bacterial superinfections. However, how human rhinovirus affects the functions of macrophages is largely unknown. Macrophages treated with HRV16 demonstrate deficient bacteria-killing activity, impaired phagolysosome biogenesis, and altered intracellular compartments. Using RNA sequencing, we identify the small GTPase ARL5b to be upregulated by the virus in primary human macrophages. Importantly, depletion of ARL5b rescues bacterial clearance and localization of endosomal markers in macrophages upon HRV16 exposure. In permissive cells, depletion of ARL5b increases the secretion of HRV16 virions. Thus, we identify ARL5b as a novel regulator of intracellular trafficking dynamics and phagolysosomal biogenesis in macrophages and as a restriction factor of HRV16 in permissive cells.**

**Keywords** Macrophages; Endosomes; Bacteria; Phagosome Maturation; Restriction Factor
**Subject Categories** Membranes & Trafficking; Microbiology, Virology & Host Pathogen Interaction

## Introduction

Human rhinovirus (HRV) is a small, non-enveloped virus with a single-stranded, positive sense RNA genome within an icosahedral capsid (Jacobs et al, 2013). HRV is known to productively infect epithelial cells (Arruda et al, 1995; Gern et al, 1996; Papi and Johnston, 1999; Sajjan et al, 2008), but there is evidence that HRV can infect monocytes and macrophages (Gern et al, 1997; Jubrail et al, 2020; Laza-Stanca et al, 2006). HRV is routinely associated with upper respiratory tract infections. However, in patients with chronic inflammatory diseases such as chronic obstructive pulmonary disease (COPD) or asthma (Yang et al, 2021), the virus can reach the lower respiratory tract and cause disease exacerbations (Gern et al, 1997; Wilkinson et al, 2006; Wilkinson et al, 2017). In addition, for a subset of patients with COPD, HRV infection can precede a secondary bacterial infection. It has been reported that HRV induces a phagocytic defect in macrophages, leading to a failure to effectively respond to secondary infections (Bellinghausen et al, 2016; Jubrail et al, 2018; Jubrail et al, 2020; Jubrail et al, 2017; Oliver et al, 2008).

Macrophages are professional phagocytes that internalize material and cell debris through surface receptors (Niedergang and Grinstein, 2018; Uribe-Querol and Rosales, 2020). Following ligand/receptor interactions, signaling cascades induce strong, local, and transient actin polymerization in addition to a remodeling of the plasma membrane, leading to the formation of a closed compartment called the phagosome. Our previous work identified the protein Arpin as an important target of the virus to downregulate phagocytic uptake (Jubrail et al, 2020).

The phagosome then matures into a highly degradative compartment termed the phagolysosome through fusion and fission steps with the endocytic machinery (Depierre et al, 2022; Fairn and Grinstein, 2012; Uribe-Querol and Rosales, 2020). Phagosomes initially acquire early endocytic markers such as the small GTPase Rab5 and its effectors among which Early Endosomal Antigen 1 (EEA1) and initiate acidification, as well as NADPH oxidase activity to generate reactive oxygen species (ROS). After a conversion from a Rab5- to a Rab7- positive compartment, phagosomes move along microtubules to fuse with late endosomes and finally lysosomes in the juxtanuclear region. Late endocytic

[1]Université Paris Cité, CNRS, Inserm, Institut Cochin, Paris 75014, France. [2]Southampton Solent University, East Park Terrace, Southampton SO14 0YN, UK. [3]Translational Science & Experimental Medicine, Research & Early Development, Respiratory and Immunology, BioPharmaceuticals R&D, AstraZeneca, Gothenburg 413 14, Sweden. [4]Institut Curie, Université PSL, CNRS, UMR144, Structure and Membrane Compartments, Paris, France. [5]Institut Curie, Université PSL, CNRS, UMR144, Cell and Tissue Imaging Facility (PICT-IBiSA), Paris, France. [6]Université Paris Est Creteil, INSERM, IMRB, Creteil 94010, France. [7]Department of Microbiology and Immunology, College of Veterinary Medicine, Cornell University, Ithaca, NY 14853, USA. [8]Immunology, Late stage Development, Respiratory and Immunology, BioPharmaceuticals R&D, AstraZeneca, Gothenburg 413 14, Sweden. [9]Research & Early Development, Respiratory and Immunology, BioPharmaceuticals R&D, AstraZeneca, Gothenburg 413 14, Sweden. [10]These authors contributed equally: Suzanne Faure-Dupuy, Jamil Jubrail. [12]These authors contributed equally: Manon Depierre, Kshanti Africano-Gomez. ✉E-mail: florence.niedergang@inserm.fr

markers such as CD63 are transiently recruited on phagosomes, followed by lysosome-associated proteins 1 and 2 (LAMP1 and LAMP2). Through fusion with lysosomes, the phagolysosomes acquire luminal proteases and other hydrolytic enzymes that participate in the degradation of the internalized material (Depierre et al, 2022; Fairn and Grinstein, 2012; Uribe-Querol and Rosales, 2020). These combined activities promote the degradation of the internalized material.

Disease exacerbations due to viral infections are thought to rely in part on the defective ability of macrophages to clear bacteria. We have recently identified that HRV alters the uptake capacities of macrophages (Jubrail et al, 2020), but whether the degradative capacities of the virus-treated macrophages are altered in these conditions is still unclear (Jubrail et al, 2017). Here, we analyze the regulation of bacterial clearance and endosome trafficking in macrophages after HRV infection. We identify the ADP Ribosylation Factor Like GTPase 5B (ARL5b), a small GTPase of the Rab, Arf and Arf-related (Arl) families, which recruit and activate the fusion machineries on specific compartments (Donaldson and Jackson, 2011; Lamber et al, 2019), as a key regulator of HRV16 infection and HRV16-mediated impairment of intracellular trafficking, phagolysosome biogenesis and thus bacterial clearance. In addition, we show that ARL5b inhibits the production of virions in cells permissive for HRV replication and therefore identified this small GTPase as a novel restriction factor in the context of HRV infection.

# Results

## Human rhinovirus 16 impairs human macrophage ability to clear respiratory bacteria

To determine if macrophages could clear internalized bacteria after the HRV16 challenge, we treated human monocyte-derived macrophages (hMDMs) with HRV16, HRV16 inactivated by a UV treatment (HRV16$^{UV}$) or mock-infected medium (MI) for 1 h at room temperature followed by overnight rest. The next day, cells were exposed to non-typeable *Haemophilus influenzae* (NTHi), *Moraxella catarrhalis, Staphylococcus aureus*, or *Pseudomonas aeruginosa* and bacterial intracellular survival was monitored 1 h after washing and incubation with antibiotics to kill extracellular bacteria. Of note, a higher multiplicity of infection (MOI) was used in HRV16-treated cells (40 bacteria per cell) to compensate for the phagocytic defect (Jubrail et al, 2020) and allow a similar number of bacteria to be taken up as in MI or HRV16$^{UV}$-treated macrophages. (Jubrail et al, 2020). Cultures were then maintained in low dose antibiotic and intracellular bacterial survival was measured over 24 h (Fig. 1A). In control conditions, the challenge of hMDMs with NTHi, *M. catarrhalis* or *P. aeruginosa* resulted in ~80% clearance after 24 h (Fig. 1B,C,E), while challenge with *S. aureus* leading to ~40% clearance after 24 h (Fig. 1D). In contrast, hMDMs challenged with HRV16 were significantly impaired in their ability to clear the four bacteria over 24 h compared to hMDMs challenged with HRV16$^{UV}$ or MI, resulting in approximately 20% clearance of NTHi, 23% clearance of *M. catarrhalis*, 7% clearance of *S. aureus*, and 14% for *P. aeruginosa* (Fig. 1B–E).

Taken together, these results demonstrate that hMDMs challenged with HRV16 are impaired in intracellular bacterial clearance and killing, suggesting modifications of phagolysosome activity and/or generation.

## Human macrophages challenged with rhinovirus 16 show reduced hydrolytic activity and reactive oxygen species production

To determine if the inability of HRV16-challenged macrophages to clear internalized bacteria could be related to phagolysosome activity, we analyzed the late steps of phagosome maturation and the luminal content of phagosomes using 3-μm beads decorated with IgG to target the Fc receptors (FcR). These beads were coupled to a fluorophore sensitive to the hydrolytic activity (DQ-BSA beads) or to the oxidative burst and the presence of reactive oxygen species (ROS) produced in the phagolysosome (dichlorodihydro-fluorescein diacetate (H$_2$DCFDA)-OxyBURST beads) as well as a calibration fluorophore (Podinovskaia et al, 2013; Yates and Russell, 2008) (Fig. 1F). The hydrolytic activity was detected after bead contact and increased with time till 120 min in control conditions (i.e., HRV16$^{UV}$ and MI conditions) (Fig. 1G). Whereas a gradual increase of hydrolytic activity over 120 min was observed in HRV16-challenged hMDMs, from 60 min to 120 min, these cells showed significantly less hydrolytic activity compared to HRV16$^{UV}$- and MI-treated cells (Fig. 1G). After 120 min, the signal declined in all conditions. The oxidative burst was detected as soon as 10 min in control conditions with a peak at 30 min and a decline of the signal by 120 min, probably due to the quenching of the fluorescein in the acidic compartment (Fig. 1H). In HRV16-treated hMDMs, ROS detection was significantly reduced at 20 and 30 min compared to MI macrophages. In HRV16-challenged hMDMs, the peak of ROS production occurred at 30 min similar to the controls. In both cases, the reduced hydrolytic and ROS activities monitored were not due to a delayed response of the macrophages (Fig. 1H).

These results demonstrate an impairment of phagolysosome activity in HRV16-challenged hMDMs.

## Human macrophages treated with human rhinovirus 16 exhibit defective phagosome maturation

To better identify where the phagosome maturation is arrested, hMDMs were allowed to internalize IgG-opsonized sheep red blood cells (SRBCs) for various time points before staining them for the early endosomal markers EEA1 (Fig. 2A,B), and Rab5 (Fig. 2C,D), the late endosomal marker CD63 (Fig. 2E,F), and Rab7 (Fig. 2G,H), and the late endosome/lysosomal marker LAMP1 (Fig. 2I,J).

EEA1 was acquired on phagosomes as early as 10 min (Figs. 2A,B and EV1A) in HRV16-treated cells as well as in control MI cells. This marker was progressively lost from phagosomes in control conditions, whereas it was still present on phagosomes of HRV16-treated macrophages at 60 min (Figs. 2A,B and EV1A). At 120 min, the percentage of EEA1$^+$ phagosomes remained stable in HRV16-treated cells, whereas it increased again in MI conditions (Fig. 2A,B). These results suggest that EEA1 recycling is happening in the MI condition. No difference in Rab5 recruitment to the phagosomes was observed at any of the time points tested, suggesting that Rab5 recruitment is not affected by HRV16 treatment (Figs. 2C,D and EV1B), while its effector EEA1 is. The late endosomal marker CD63 was enriched on phagosomes at 30 min post SRBCs exposure in control conditions and less present at 60 and 120 min (Fig. 2E,F). By contrast, it remained associated with the phagosomes at the same level at 30, 60, and 120 min in

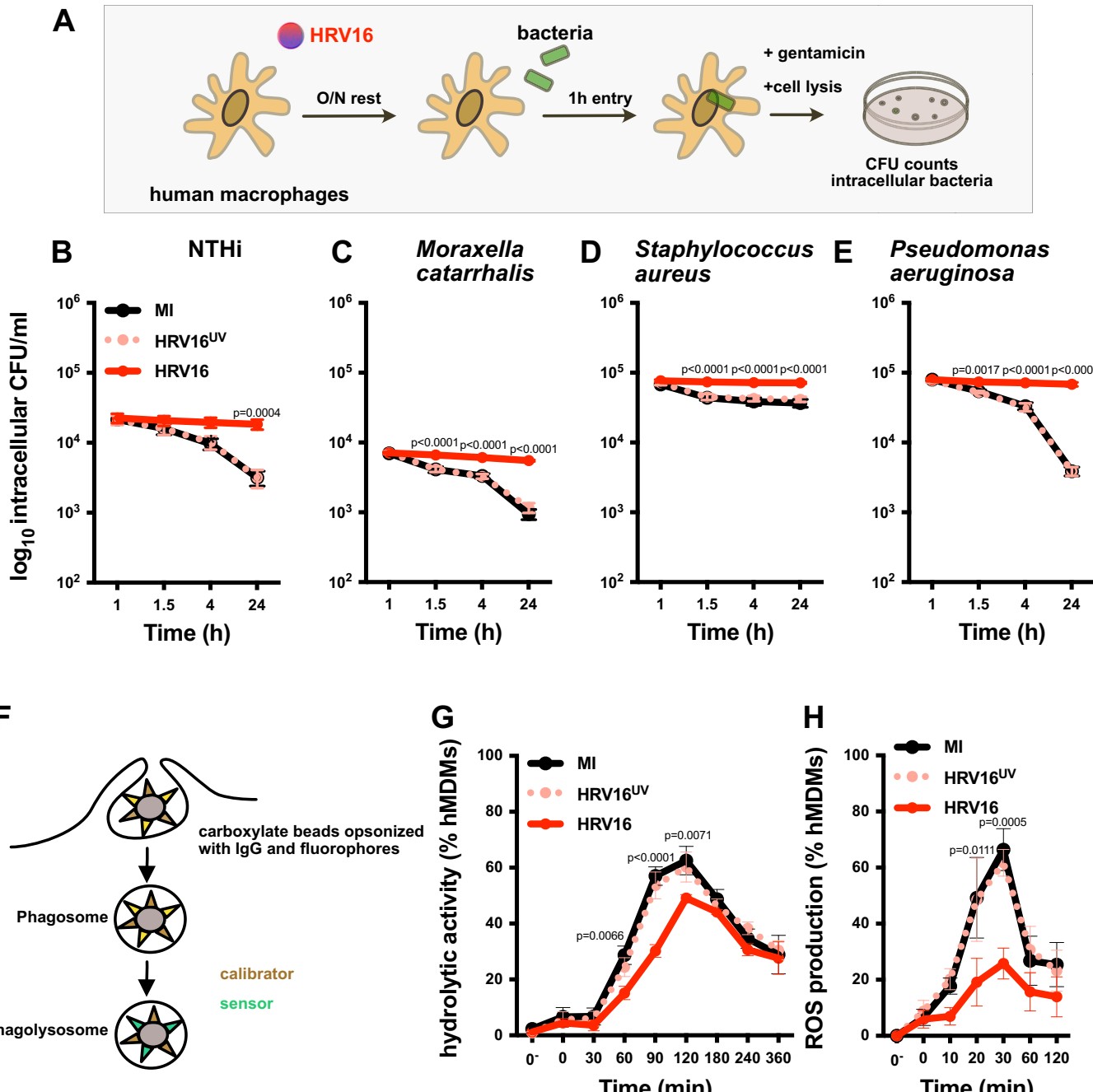

**Figure 1. HRV16 impairs bacterial clearance and phagolysosome activity by macrophages.**

(A–E) hMDMs were challenged with HRV16, HRV16^UV, or MI. After O/N rest, cells were exposed to bacteria to perform a gentamicin protection assay and assess bacterial survival at the indicated time points. (A) Schematic representation of the experiment. (B–E) Bacteria were either infected with (B) NTHi, (C) *Moraxella catarrhalis*, (D) *Staphylococcus aureus*, or (E) *Pseudomonas aeruginosa*, and bacteria survival was determined. (F) Schematic representation of bead assay to measure ROS production and protease activity within phagosomes by flow cytometry. (G, H) hMDMs were challenged with HRV16, HRV16^UV or MI and then exposed to IgG-opsonized beads to measure (G) ROS or (H) protease production in mature phagolysosomes. Data information: (B–E, G, H) Points represent the mean of three biological replicates of independent experiments +/− SEM. Two-way ANOVA with Dunnett's post test statistical analysis was performed. Source data are available online for this figure.

HRV16-treated macrophages (Fig. 2E,F). Rab7 was present on phagosomes at 30 min but its kinetics of disappearance from phagosomes varied with donors (Fig. 2G,H). Although not significantly different, a tendency for an increase of Rab7 on phagosomes was observed at 30 min in HRV16-treated cells

compared with MI controls in each of the experiments (Fig. 2G,H). Finally, there was significantly less recruitment of LAMP1 on phagosomes in HRV16-treated macrophages compared to control cells at 30 and 60 min, and a non-significant trend to lower recruitment at 120 min (Fig. 2I,J).

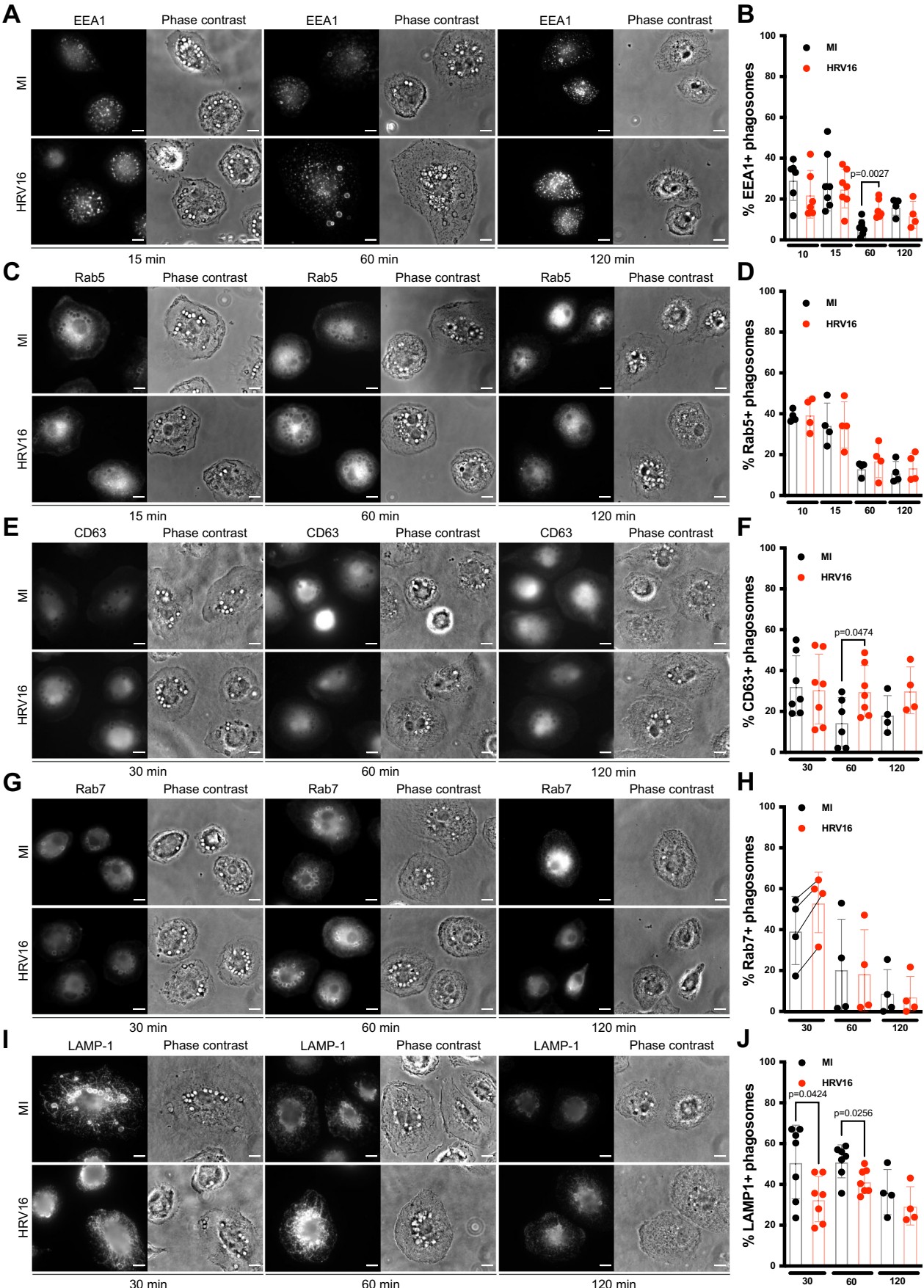

◄ **Figure 2. HRV16 impairs phagosome maturation in macrophages.**

(A–J) hMDMs were challenged with HRV16 or MI and then exposed to IgG-opsonized sheep red blood cells for the indicated time and stained with either (**A, B**) EEA1, (**C, D**) Rab5, (**E, F**) CD63, (**G, H**) Rab7 or (**I, J**) LAMP1. (**A, C, E, G, I**) Representative images of (**A**) EEA1, (**C**) Rab5, (**E**) CD63, (**G**) Rab7, (**I**) LAMP1 staining at the indicated time points in MI (upper row) and HRV16-treated cells (lower row). (**B, D, F, H, J**) Percentage of (**B**) EEA1, (**D**) Rab5, (**F**) CD63, (**H**) Rab7, or (**J**) LAMP1-positive phagosomes at the indicated times. Data information: Bars graph represents the mean of at least four biological replicates of independent experiments +/− SD. Unpaired student $t$ test statistical analysis was performed for each time point. Scale bar represents 10 µm. Source data are available online for this figure.

Together, these results indicate that the stalled phagosomes have intermediate phenotypes with normal presence of Rab5, early recruitment of Rab7, prolonged EEA1, CD63, and lower recruitment of Lamp1. There is some recovery with time, as the differences are no more significant, but there is still a tendency for the phagosomes to keep higher CD63 and lower Lamp1 recruitment in macrophages treated with HRV16 as compared with the control cells, indicating a perturbed biogenesis of phagolysosomes.

## Human rhinovirus 16 impairs the expression of EEA1 and CD63 in macrophages

Next, we investigated whether HRV16 challenge of hMDMs would impair the localization or the expression of endocytic markers, which could lead to a defective recruitment around phagosomes. For this, following HRV16 challenge or mock infection, we first stained hMDMs for EEA1. We observed that the EEA1 staining was uniformly distributed throughout the cytosol in MI hMDMs (Fig. 3A, left panels). By contrast, HRV16-challenged hMDMs exhibited a more intense EEA1 staining throughout the cell as well as more heterogeneous vesicles than in the MI control (Fig. 3A, right panels). Quantification of the fluorescence intensities confirmed that, relative to MI hMDMs, the intensity of EEA1 staining in HRV16-challenged hMDMs was significantly higher (Fig. 3B). In addition, there was no change in the number of EEA1-positive vesicles in HRV16-challenged hMDMs (Fig. 3C), but the total amount of EEA1 protein was increased in HRV16-challenged hMDMs (Fig. 3D,E).

Furthermore, we observed that HRV16-challenged hMDMs had a more heterogeneous localization of CD63 than MI cells (Fig. 3F), with increased staining accumulated at the periphery and close to the plasma membrane (Fig. 3F, right panel). CD63-positive vesicles also appeared to have a higher intensity. Image quantification revealed that CD63-associated fluorescence intensity was significantly higher in HRV16-treated hMDMs compared to MI cells (Fig. 3G). Furthermore, the number of CD63-positive late endosomes was significantly higher in HRV16-challenged hMDMs compared with MI hMDMs (Fig. 3H). In addition, the surface expression of CD63 was assessed by flow cytometry to analyze the general trafficking of the protein. Indeed, CD63 traffics to the plasma membrane before being addressed to late endosomes. In HRV16-challenged hMDMs, the surface level of CD63 was two times higher than in MI macrophages (Fig. 3I).

These results led us to investigate the ultrastructure of the intracellular compartments of HRV16-treated hMDMs by conventional electron microscopy (see "Methods"). Although fluorescent intensities associated with EEA1 and CD63 were increased in virus-treated cells, the overall morphology of endosomal compartments, and especially of early endosomes, were not altered compared to control cells (Fig. 3J, arrows). However, we noticed during the analyses that the ultrastructure of the Golgi apparatus and of the Trans Golgi Network (TGN) displayed changes. In non-infected cells, these compartments presented an expected structure, illustrated by visible stacked Golgi

cisternae (Fig. 3J, top panels, arrowheads). However, in HRV16-treated hMDMs, the Golgi apparatus/TGN were less compact, with less well-structured cisternae associated with numerous tubulo-vesicular structures (Fig. 3J, bottom panels, arrowheads). In addition, we performed immunofluorescence staining of the HRV16-treated macrophages and observed an increase in the fluorescent signal for GM130 and a decrease in the TGN46 signal in virus-treated cells as compared with control cells (Fig. 3K,L), suggesting that the morphological and molecular integrity of the Golgi apparatus is affected following HRV16 challenge.

Taken together, these results highlight that the virus induces modifications in intracellular trafficking, probably both at the level of anterograde and retrograde trafficking, with altered expression and localization of endosomal compartments, which could contribute to the perturbation in their recruitment on phagosomes.

## Transcriptomic analysis reveals that human rhinovirus 16 upregulates ARL5b in macrophages

To identify potential host cell candidates that HRV16 could affect in macrophages to drive the perturbations observed, we performed an RNA sequencing on HRV16-treated *versus* HRV16$^{UV}$-treated and MI-treated hMDMs. We identified 2067 differentially expressed genes induced by HRV16 and 2471 induced by HRV16$^{UV}$ as compared to MI hMDMs with a common overlap of 1510 genes (false discovery rate <0.05) (Fig. 4A; Datasets EV1 and EV2). Pathways activated in the HRV16 as compared to MI conditions are linked to immune responses and cell signaling (Fig. EV2A). When we compared the fold changes induced by HRV16 or HRV16$^{UV}$, we identified a group of genes for which the responses to the two treatments differed greatly (Fig. 4B). A total number of 160 genes showed a larger variation from the normal spread (Fig. 4B blue; Dataset EV3). Analysis of these 160 genes highlighted cell activation and induction of signaling pathways, but there was no major highlight on intracellular trafficking (Fig. EV2B). Functional and disease term enrichment analysis for the HRV16 vs MI and HRV16-UV vs MI responses for the 160 genes are shown in Fig. EV2B and Dataset EV3, where an absolute z-score of at least 2 is considered significant. When clustering these 160 genes, clusters 2 and 3 showed a response unique to HRV16 while clusters 1 and 4 had a response unique to HRV16$^{UV}$ (Fig. 4C). Cluster 2 contained 33 genes more highly expressed in HRV16-challenged hMDMs than in control conditions. One of the most consistently and significantly upregulated genes in the HRV16 condition was the small GTPase ARL5b (Fig. 4D). ARL5b was reported to control anterograde and retrograde trafficking from and toward the Golgi apparatus (Houghton et al, 2012). As we observed that the endocytic compartments and the Golgi apparatus were impacted by HRV16 (Fig. 3J–L), we hypothesized that ARL5b could be involved in the modifications observed. The threefold upregulation of ARL5b transcripts was confirmed by RT-qPCR in HRV16-treated

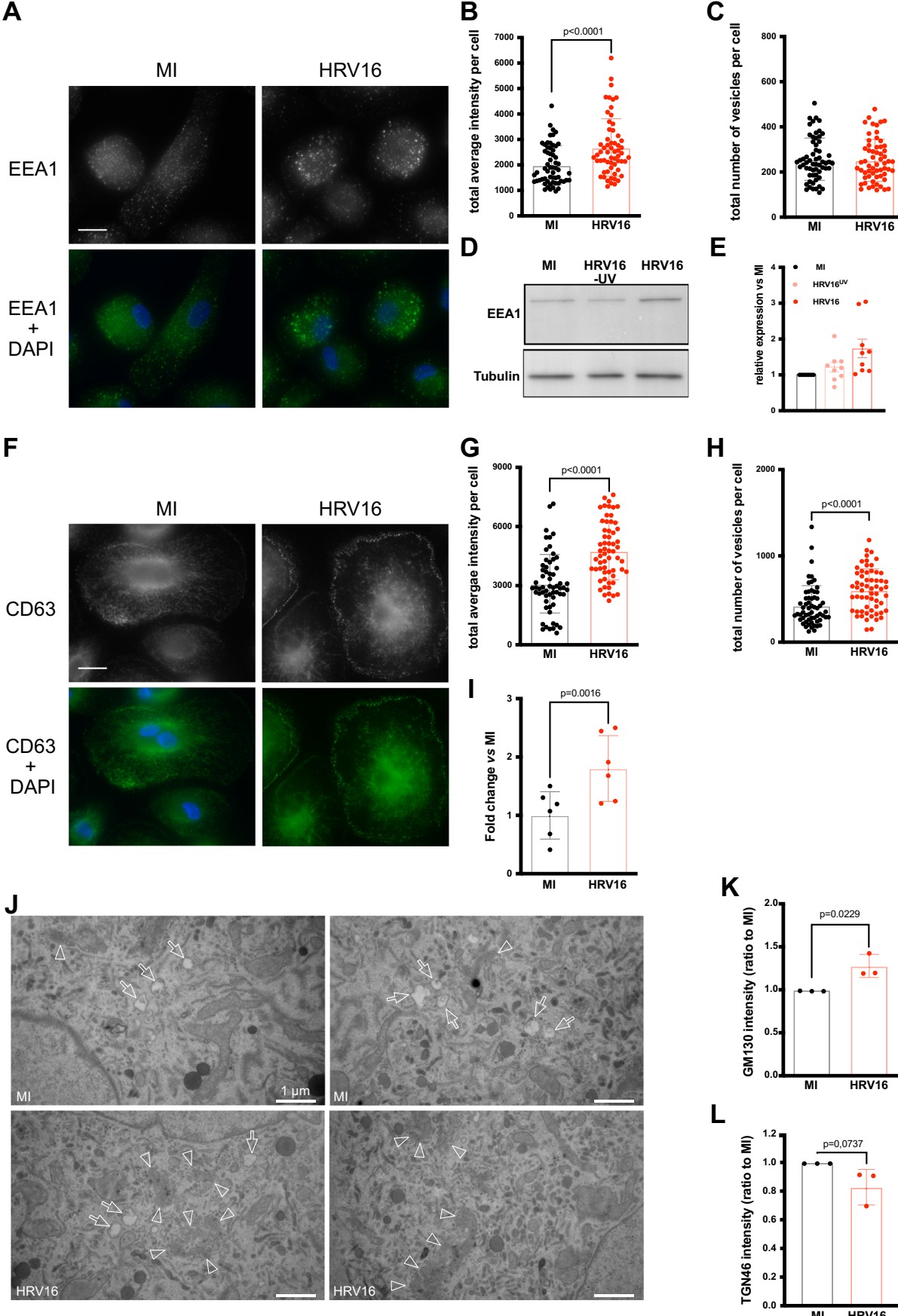

Figure 3.   HRV16 impairs endosomal markers localization and expression, as well as the Golgi apparatus morphology in macrophages.

(A–C) hMDMs were challenged with HRV16 or MI and were then stained for EEA1. (A) Representative image of EEA1 staining in MI (left) or HRV16-treated cells (right) with (lower row) or without DAPI (upper row). (B) Quantification of the intensity of EEA1 staining per cell in 60 randomly chosen cells. (C) Quantification of the total number of EEA1-positive endosomes per cell in 60 randomly chosen cells. (D, E) hMDMs were challenged with HRV16, HRV16$^{UV}$ or MI and total protein was lysed for western blot. (D) Representative western blot with anti-EEA1 and anti-Tubulin as a loading control. (E) Quantification of EEA1 protein normalized to tubulin and presented as ratio to the MI control, $n = 9$. (F–H) hMDMs were challenged with HRV16 or MI and were then stained for CD63. (F) Representative image of CD63 staining in MI (left) or HRV16-treated cells (right), with (lower row) or without DAPI (upper row). (G) Quantification of the intensity of CD63 staining per cell in 60 randomly chosen cells. (H) Quantification of the total number of CD63-positive endosomes per cell in 60 randomly chosen cells. (I) hMDMs were challenged with HRV16 or MI and then surface localized CD63 was analyzed by flow cytometry, $n = 8$. (J) Conventional EM micrographs of ultrathin sections of hMDMs treated or not with HRV16 (bottom or top panels, respectively). Arrows point to early endosomes with no obvious morphological alterations. Arrowheads point to Golgi apparatus/TGN with perturbed morphology in HRV16-treated cells (bottom panels) compared to control (top panels). (K, L) hMDMs were challenged with HRV16 or MI. Cells were stained for (K) GM130 or (L) TGN46 by immunofluorescence. Data show the quantification of the average intensity per cell of at least 30 randomly chosen cells. Data information: (A, F) Scale bar represents 15 μm. (J) Scale bars represent 1 μm. (B, C, E, G–I, K, L) Bar graphs represent the mean of at least three biological replicates of independent experiments $+/-$ SD. Paired student $t$ test statistical analysis was performed. Source data are available online for this figure.

compared to MI-treated hMDMs (Fig. 4E). This increase was also detected at the protein level (Fig. 4F,G).

The role of ARL5b in macrophages remains unknown. We therefore decided to investigate further its function in HRV16-treated macrophages.

## ARL5b depletion prevents HRV16-mediated endosomal defects and restores bacterial clearance in macrophages

To determine if decreasing ARL5b expression in hMDMs could compensate for the HRV16-induced phenotypes previously observed (Figs. 1B–E and 3), ARL5b was depleted using siRNAs. hMDMs were treated with two siRNAs sequences against ARL5b (siARL5b.1 and siARL5b.2) or a control siRNA targeting Luciferase (siLuc) for 24 h and then challenged with HRV16 or with MI medium. We confirmed that ARL5b expression was significantly reduced by both siRNAs, ranging from 40% up to 60% reduction compared to their corresponding controls (Fig. 5A,B). We then further analyzed if ARL5b depletion could rescue HRV16-induced phenotypes in hMDMs (Fig. 5C–G).

We observed that hMDMs treated with siLuc and HRV16-challenged showed a significant increase in the intensity of EEA1 staining relative to MI controls (Fig. 5C,D), in agreement with the results presented in Fig. 3. By contrast, treatment with siARL5b.1 or siARL5b.2 blocked the HRV-dependent increase of EEA1 staining (Fig. 5C, middle and right panels and D). Importantly, the depletion of ARL5b did not affect the EEA1 intensity in MI hMDMs (Fig. 5C,D).

In agreement with our previous results (Fig. 3), a twofold increase in surface localized CD63 in HRV16-treated hMDMs compared to MI was observed in siLuc-treated macrophages (Fig. 5E). Interestingly, treatment with siARL5b.1 or siARL5b.2 blocked the HRV-dependent increase of CD63 surface expression levels (Fig. 5E).

Moreover, whereas treatment of hMDMs with siLuc before HRV16 challenge did not alter the phenotype and led to impaired NTHi clearance, with significant intracellular persistence at 2 h and 4 h compared to MI hMDMs (Fig. 5F), treatment with siARL5b.1 or siARL5b.2 improved bacterial clearance (Fig. 5F).

Finally, we observed that hMDMs treated with siLuc and HRV16-challenged showed a significant decrease of the percentage of LAMP1$^+$ phagosomes in agreement with the results of Fig. 2I,J (Figs. 5G and EV3). In comparison, treatment with siARL5b.1 or siARL5b.2 restored the Lamp1 recruitment on phagosomes in

HRV16-treated cells (Figs. 5G and EV3), indicating a rescue of phagosome maturation upon ARL5b depletion.

Taken together, our results demonstrate in human macrophages that the HRV16-induced increase of the small GTPase ARL5b drives endosomal perturbations and leads to an inefficient intracellular bacterial clearance, and hence, potentially, bacterial persistence.

## ARL5b acts as a restriction factor of HRV16 in permissive HeLa Ohio cells

Given the role of ARL5b in Golgi trafficking, we then investigated if the increase of ARL5b expression could play a direct role on HRV16 replication and propagation. HRV is known to replicate in epithelial cells. Therefore, to address the effect of ARL5b on HRV16 replication, we first used a model permissive to its replication, the HeLa Ohio cell line.

HeLa Ohio were transfected with plasmids expressing WT ARL5b (pARL5b WT) or an inactive ARL5b mutant (pARL5b T30N) or the corresponding control plasmids (pCtrl WT or pCtrl T30N, respectively) (Fig. 6A). Efficiency of the transfection and ARL5b overexpression was confirmed by RT-qPCR (Fig. 6B). No effect of ARL5b overexpression (WT or T30N mutant) was observed on HRV mRNAs levels, indicating that ARL5b has no direct role on HRV16 replication (Fig. 6C). To assess if ARL5b might play a role on HRV16 propagation, supernatants from HeLa Ohio transfected with the different plasmid and then infected were collected and used to reinfect non-transfected HeLa Ohio (Fig. 6A). Infection was then monitored through a TCID50 assay. Interestingly, supernatants from the cells overexpressing WT ARL5b ("pARL5b WT" condition) showed a lower TCID50 than the one infected with the corresponding control ("pCtrl WT" condition) (Fig. 6D). This highlights that in permissive cells, overexpression of ARL5b limits the secretion of HRV16 virions and thus the propagation of the infection. No difference was observed between "pARL5b T30N" and "pCtrl T30N" conditions (Fig. 6D), indicating that an active ARL5b is needed to obtain this phenotype.

Subsequently, to confirm that ARL5b inhibits HRV16 propagation but not its replication, HeLa Ohio were transfected with siRNAs against ARL5b (siARL5b.1 and siARL5b.2) or a control siRNA targeting Luciferase (siLuc) for 24 h and then infected with HRV16 or MI. ARL5b depletion was confirmed by RT-qPCR (Fig. 6E). No impact of ARL5b depletion was observed on HRV mRNAs levels, confirming that ARL5b plays no role on HRV16 replication (Fig. 6F). Supernatants from HRV16-infected

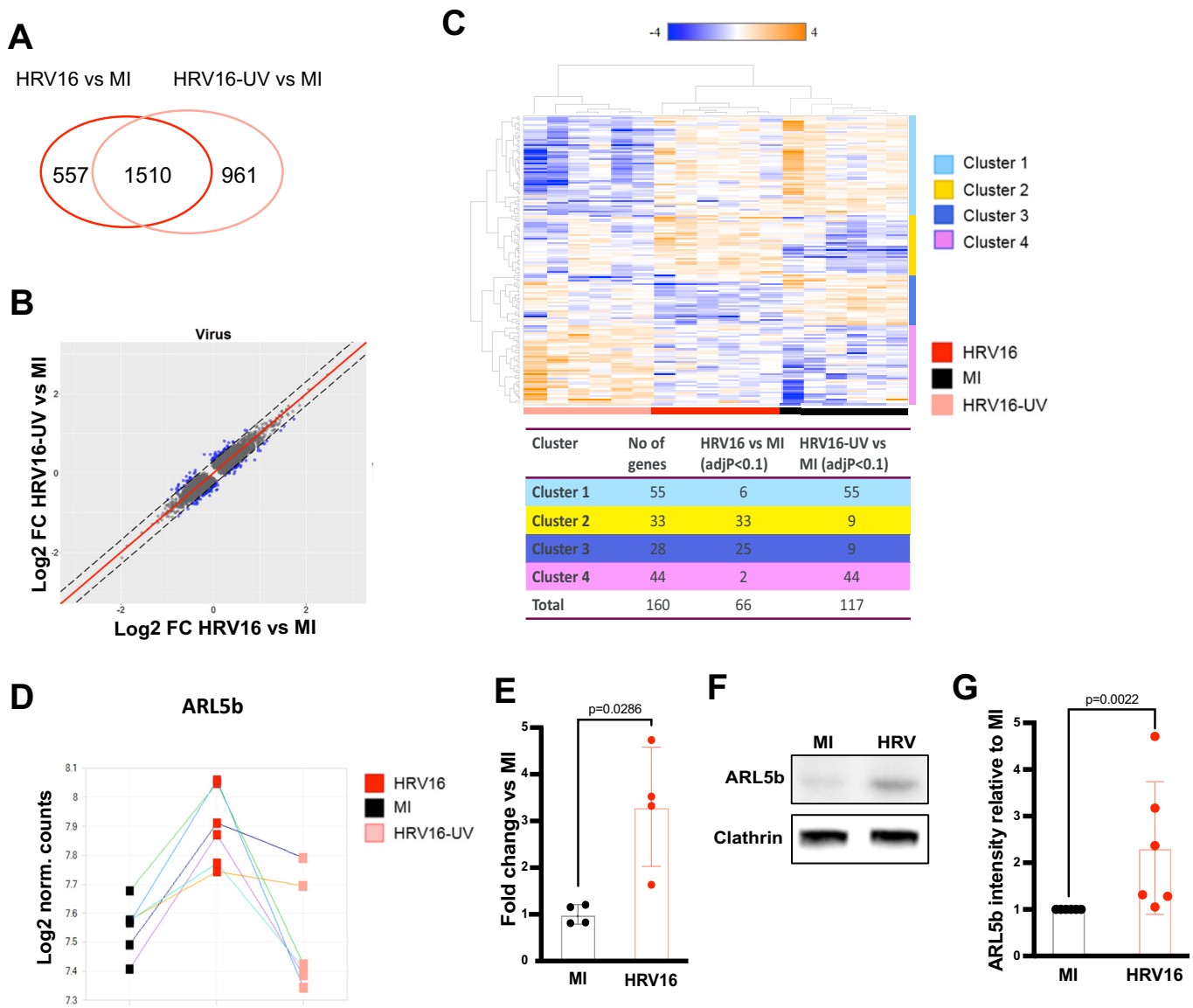

**Figure 4. HRV16 increases ARL5b expression in macrophages.**

(A–D) hMDMs were challenged with HRV16, HRV16UV, or MI and characterized by RNA sequencing, n = 6. (A) Venn diagram of the results. The dark red circle represents the genes significantly regulated in the HRV16 vs MI analysis, and the light red circle represents the genes significantly deregulated in the HRV16UV vs MI analysis (FDR < 0.05). (B) By comparing the fold changes induced by HRV16 and HRV16UV, respectively (FDR < 0.1), a set of genes could be identified where the two responses differ to a larger extent. If the two responses are equal, they will fall on the diagonal line (red). The further off the diagonal, the more different the responses. Hundred and sixty genes fall outside of the +/− 2 SD cut-off (dotted blue lines). (C) Hierarchical clustering of the 160 genes with the most deviating responses to HRV16 and HRV16UV revealed four different clusters. (D) ARL5b is upregulated by the live HRV16 virus (FC: 1.3, FDR: 0.0091) but not by the inactivated virus (HRV16UV) (FC: −1.0, FDR: 0.97). Colored lines indicate paired samples by donor. (E–G) hMDMs were challenged with HRV16 or MI. (E) ARL5b was analyzed by RT-qPCR. (F, G) ARL5b expression was analyzed by western blot. (F) Representative western blot with anti-ARL5b and anti-Clathrin as a loading control. (G) Quantification of ARL5b expression by ImageJ. Data information: (E) Bar graph represents the mean fold change +/− SD of HRV16 vs MI normalized to 18 S rRNA housekeeping gene of four independent experiments. Mann–Whitney U test was performed. (G) The bar graph represents the mean +/− SD of six biological replicates of independent experiments. Unpaired student t test statistical analysis was performed. Source data are available online for this figure.

ARL5b-depleted cells showed a higher TCID50 then the condition (Fig. 6G). Therefore, depletion of ARL5b in HeLa Ohio cells increases HRV16 virions secretion and viral propagation.

Given that ARL5b increase has a negative effect on HRV16 virions secretion, we wondered if ARL5b would be differentially regulated in permissive cells. Upon HRV16 infection, a decrease of

ARL5b mRNA expression was observed as early as 4 h post infection and onwards (Fig. 6H). This decrease was concomitant with an increase in viral mRNAs, suggesting that viral replication inhibits ARL5b expression (Fig. 6I).

To assess if these results were representative of the effect of HRV16 on epithelial cells, we performed similar experiments on an

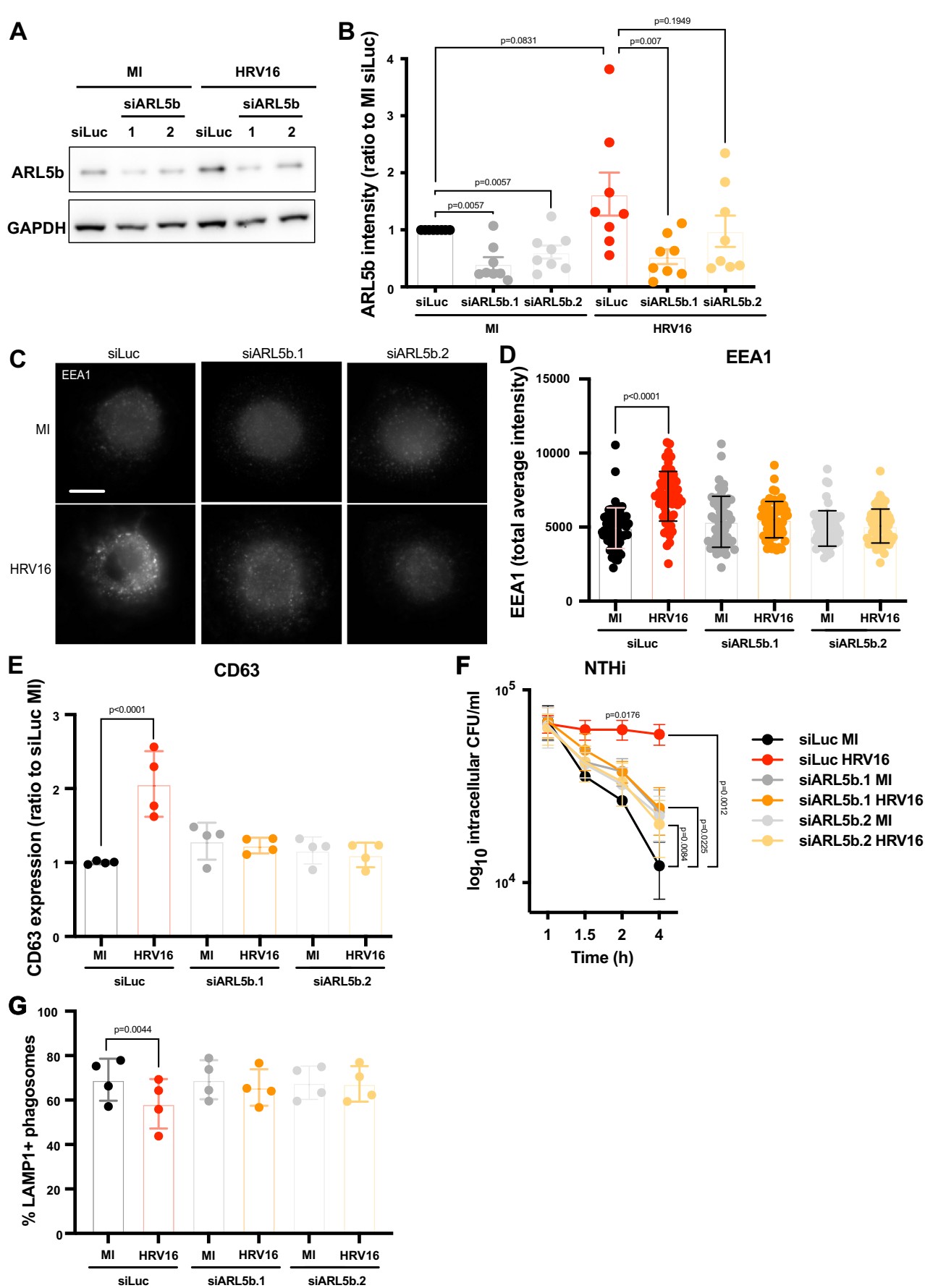

**Figure 5.  ARL5b depletion rescues the HRV16-induced modifications in macrophages.**

(A–G) hMDMs were transfected with siRNA against luciferase (siLuc, control) or ARL5b (siARL5b.1 and siARL5b.2) and then challenged with HRV16 or MI. (A, B) Cells were lysed to analyze the protein content by western blot. (A) Representative western blot with anti-ARL5b and anti-GAPDH as a loading control. (B) Quantification of the ARL5b band intensity relative to GAPDH and normalized to the control MI siLuc condition. (C, D) Cells were stained to detect EEA1. (C) Representative images of EEA1. (D) Quantification of the intensity of EEA1 staining in 60 random cells. (E) CD63 surface staining was analyzed by flow cytometry. (F) After O/N rest, cells were exposed to bacteria (NTHi) to perform a gentamicin protection assay and assess bacterial survival. (G) Cells were exposed to IgG-opsonized sheep red blood cells for 60 min and stained with LAMP1. Graphs represent the percentage of LAMP1-positive phagosomes. Data information: (C) Scale bar represents 15 μm. (B, D–F) Graphs represent the mean +/- SEM of at least three biological replicates of independent experiments. (B) One sample t test, (D, E) One-way ANOVA with Bonferonni post test or (F) two-way ANOVA with Dunnett's post test statistical analysis were performed. (G) Bar graphs represent the mean of four biological replicates of independent experiments +/− SD. One-way ANOVA statistical analysis was performed for each time point. Source data are available online for this figure.

epithelial cell line, the BEAS-2B. We showed that BEAS-2B, which are non-transformed lung epithelial cells, are not permissive to HRV16 infection as shown by a constant decrease on HRV16 RNA post infection (Fig. 6J). Interestingly, in these epithelial cells, ARL5b expression was not modified from 0 h to 96 h post infection and increased at 120 h post infection (Fig. 6K).

Therefore, the regulation of ARL5b is not identical in all epithelial cells and seems to be correlated with the permissiveness of the cells allowing viral replication, in which it acts as a restriction factor.

## Discussion

We show that HRV16 impairs intracellular bacterial clearance in macrophages, by causing defective phagosome maturation and degradation. We reveal that the small GTPase ARL5b is upregulated by the virus in macrophages and plays a crucial role in this perturbation by regulating the subcellular localization and recruitment of endocytic compartments (Fig. 7). We also show that ARL5b inhibits HRV16 secretion in permissive cells, and therefore we identified ARL5b as a novel restriction factor of HRV16 infection in this context (Fig. 7). Therefore, ARL5b plays a dual role during HRV16 infection.

HRV16-treated macrophages could not efficiently clear internalized bacteria. Importantly, the defective clearance was not restricted to a particular bacterial species but was observed for both Gram-positive and Gram-negative bacteria (Fig. 1). These results suggested general deficiencies in degradative capacities in the phagolysosomes of HRV16-treated macrophages. Intracellular killing of bacteria within mature phagolysosomes in macrophages requires both the generation of antimicrobial molecules and appropriate phagolysosomal maturation and acidification (Depierre et al, 2022; Flannagan et al, 2009). Our results demonstrated that HRV16 significantly decreased ROS production and hydrolytic activity of macrophages after receptor-mediated phagocytosis. These findings could offer some explanation for the defective bacterial clearance activity of macrophages upon HRV16 exposure.

Phagosome formation and maturation is a complex process depending on various cellular activities (Depierre et al, 2022; Mularski and Niedergang, 2017). The phagosome maturation to the phagolysosome is characterized by the subsequent recruitment of endosomal proteins, including Rab5 and its effector EEA1, late endosomal proteins, such as CD63 or Rab7, and lysosomal proteins like LAMP1 and LAMP2. Membrane recycling allows the resolution of the phagosome and lysosome recovery (Lancaster et al, 2021). We found that EEA1, CD63, Rab7, and LAMP1 were not recruited normally to phagosomes in HRV16-treated macrophages,

while Rab5 was, leading to a subset of phagosomes stalled between early and late steps of maturation, which could explain the defective degradative capacities of these compartments. Importantly, there were also alterations of the Golgi apparatus, endosomal and lysosomal markers, as well as modifications in their expression at the protein level. Using electron microscopy, we did not observe changes in the morphology of the endosomal compartments. However, we noticed that the ultrastructure of the Golgi apparatus and the TGN was altered. Together, these data indicate broad changes in the intracellular endosomal and/or secretion trafficking, which were due to the presence of live HRV16, as we did not observe the same changes in macrophages challenged with UV-inactivated HRV16.

The unbiased approach using RNA sequencing of HRV-infected macrophages allowed us to identify one potential gene related to transport from and to the TGN (Houghton et al, 2012), the small GTPase ARL5b, which was upregulated after infection. Depletion of ARL5b rescued all the observed HRV16-mediated phenotypes (i.e., phagosome maturation and bacterial clearance), which highlights ARL5b as a central element of HRV16 block of bacterial clearance by perturbation of the intracellular trafficking in macrophages.

Since ARL5b controls traffic between endosomes and the Trans Golgi Network (Houghton et al, 2012), we then hypothesized that ARL5b overexpression could participate in the membrane fluxes induced by the viral infection. Contrary to what was observed in macrophages, when we analyzed the role of ARL5b in HRV16-permissive cells (i.e., HeLa Ohio), we observed that ARL5b is a negative regulator of HRV16 virions secretion, highlighting that ARL5b is a restriction factor for HRV16 in these cells.

The differences of HRV16-induced modification of ARL5b expression between macrophages and HeLa Ohio cells could be due to differences in immune responses or in viral replication in these cells. ARL5b has been identified to be an interferon-stimulated gene (Boppana et al, 2013). Thus, we can hypothesize that in macrophages, which have a strong type I interferon response, an induction of ARL5b would be observed. In HeLa Ohio, which do not produce interferon beta, ARL5b levels would remain stable upon infection. However, in HeLa Ohio cells, ARL5b expression is decreased by the infection. This indicates that HRV16 infection leads to inhibition of ARL5b expression. Therefore, we can also hypothesize that the difference of ARL5b regulation between these cells could be linked to the permissiveness of the cells to the infection. This hypothesis is strengthened by our results in the non-permissive epithelial cell line BEAS-2B, in which ARL5b expression was increased at late time points post infection and in which HRV16 does not replicate.

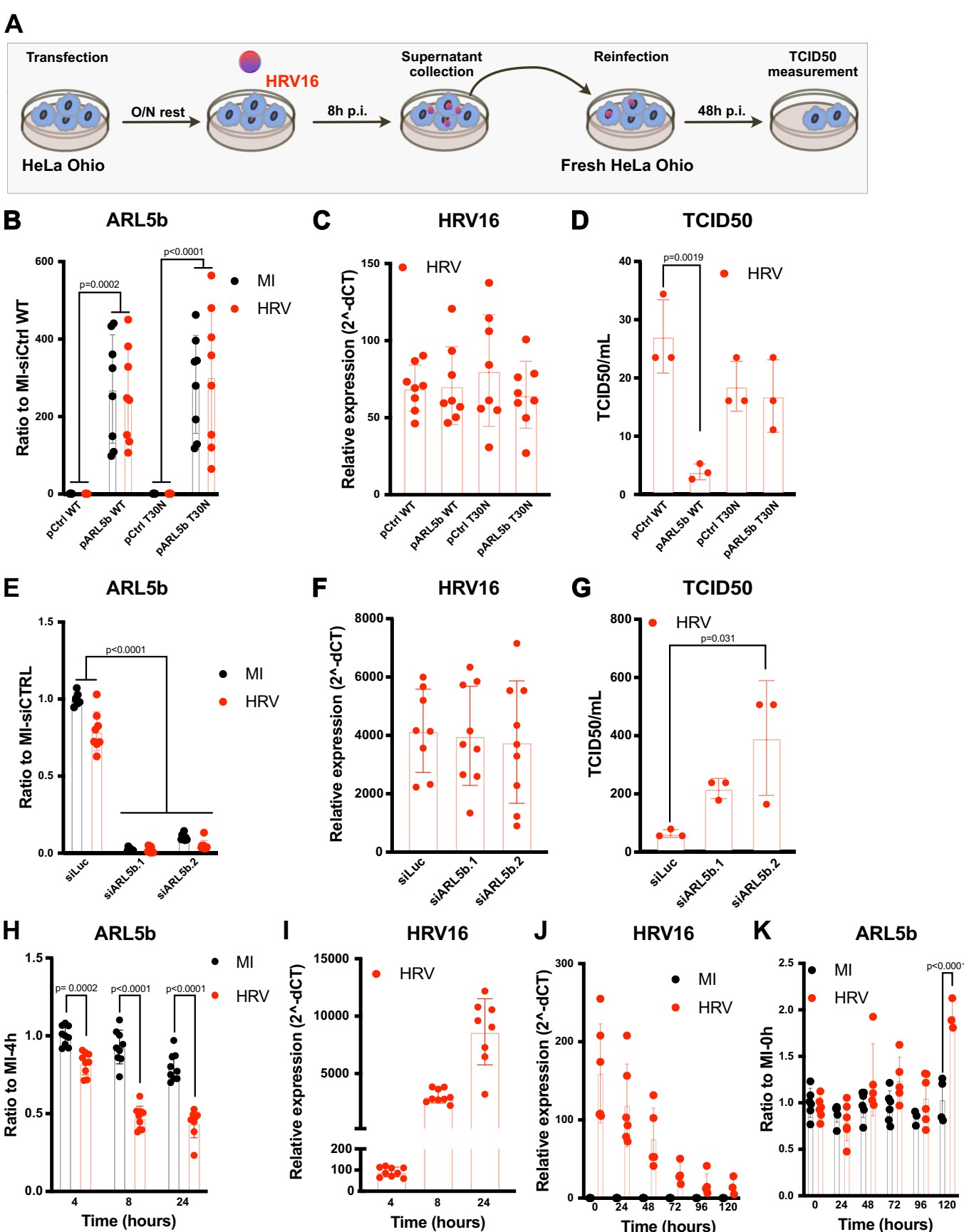

**Figure 6. ARL5b inhibits HRV16 virion secretion in permissive cells.**

(A) Schematic representation of the experiments presented in (B–G). (B–D) HeLa Ohio were transfected with a plasmid expressing a WT ARL5b coupled to mCherry (pARL5b WT) or expressing an inactive mutant of ARL5b coupled to GFP (pARL5b T30N) as well as corresponding control plasmids expressing similar fluorescent proteins (pCtrl WT and pCtrl T30N, respectively). Twenty-four hours post transfection, cells were infected with HRV16 or MI. Supernatants were collected after 8 h and cells were lysed for RNA analysis. (B, C) mRNA were extracted and (B) ARL5b and (C) HRV mRNAs were analyzed by RT-qPCR. (D) TCID50 was determined on the collected supernatants. (E–G) HeLa Ohio were transfected with siRNA against luciferase (siLuc, control) or ARL5b (siARL5b.1 and siARL5b.2). 24 h post transfection, cells were infected with HRV16 or mock-infected. Supernatants were collected after 8 h, and cells were lysed. (E, F) mRNA were extracted and (E) ARL5b and (F) HRV mRNAs were analyzed by RT-qPCR. (G) TCID50 was determined on the collected supernatants. (H, I) HeLa Ohio infected with HRV16 or MI. Cells were lysed at the indicated time points. mRNA were extracted and (H) ARL5b and (I) HRV mRNAs were analyzed by RT-qPCR. (J, K) BEAS-2B were infected with HRV16 or MI. Cells were lysed at the indicated time points. mRNA were extracted and (J) HRV and (K) ARL5b mRNAs were analyzed by RT-qPCR. Data information: (B–K) Bar graphs represent the mean +/− SD of at least three biological replicates of independent experiments performed in triplicates. (B, E) Two-way ANOVA, (D, G) one-way ANOVA, or (H, K) student $t$ test statistical analysis were performed. Source data are available online for this figure.

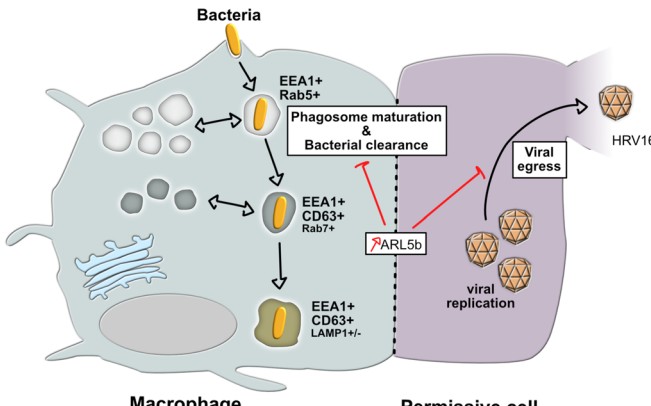

**Figure 7. Graphical abstract.**

Summary of the results obtained in this study. Exposure to HRV upregulates ARL5b expression in macrophages, which leads to impairment of the intracellular trafficking and defect in phagosome maturation. This translates into stalled EEA1[+] CD63[+] LAMP1[+/−] phagosomes and a defective bacteria clearance. Moreover, ARL5b upregulation in permissive cells inhibits viral egress.

HRV frequently exacerbates chronic inflammatory lung diseases, including asthma and COPD (Gern et al, 1997; Wilkinson et al, 2006; Wilkinson et al, 2017). Our study provides a novel analysis of how HRV impairs phagocytic processes downstream of uptake in primary macrophages. The present study thus extends previous reports showing that HRV-treated macrophages are less efficient in capturing bacteria (Jubrail et al, 2020) and could not mount an efficient cytokine response to bacterial products (Finney et al, 2019; Jubrail et al, 2018; Oliver et al, 2008). With the present study, all the steps of phagocytosis, from bacterial capture to degradation and cell activation, are impaired upon HRV16 challenge of macrophages, which could explain why patients show bacterial outgrowths post-HRV exposure (Singanayagam et al, 2019; Wilkinson et al, 2006; Wilkinson et al, 2017). Importantly, in our previous work and in this study, we have identified novel host cell factors implicated in two of these steps, namely the actin regulator Arpin, which is crucial for efficient phagocytosis and bacterial uptake (Jubrail et al, 2020), and the ARL5b which is induced by the infection and prevents phagosome maturation and bacterial clearance. The identification of the mechanistic pathways involved and the host factors targeted might

open new avenues for the development of strategies to improve the phagocytic and clearance activities of macrophages in chronic respiratory diseases.

## Methods

### Monocytes-derived macrophages differentiation

Human peripheral blood mononuclear cells (PBMCs) were isolated from whole blood of healthy donors (Etablissement Français du Sang Ile-de-France, Site Trinité, Inserm agreement #15/EFS/012 and #18/EFS/030) ensuring that all donors gave a written informed consent and providing anonymized samples, with no gender discrimination and no genetic modification) by density gradient sedimentation using Ficoll-Paque (GE Healthcare). This was followed by adhesion on plastic at 37 °C for 2 h and culture in the presence of adhesion medium (RPMI 1640 (Life Technologies) supplemented with 100 μg/ml streptomycin/penicillin and 2 mM L-glutamine (Invitrogen/Gibco)). Then, the adhered cells were washed once with warm adhesion medium and left to rest in macrophage culture medium (adhesion medium supplemented with 10% FCS (Eurobio)) at 37 °C. The next day, cultures were washed with adhesion medium and then supplemented every 2 days with fresh macrophage culture medium. The adherent monocytes were left to differentiate at 37 °C into macrophages as described previously (Jubrail et al, 2020) and used after 10 days.

### HeLa Ohio culture

HeLa Ohio cells were purchased from the European Collection of Authenticated Cell Cultures (ECACC, 84121901) and were cultured at 37 °C in DMEM GlutaMax containing 25 mM D-glucose (Life Technologies) supplemented with 10% FCS, 100 μg/ml streptomycin/penicillin and 2 mM L-glutamine. They were passaged every 3 days. Cells were tested negative for *Mycoplasma* contamination. Cells were not recently authenticated.

### BEAS-2B culture

BEAS-2B cells were purchased from the ATCC (CRL3588) by Dr. Maha Zohra Ladjemi and kindly gifted to us. Cells were cultured at 37 °C in F12K with glutamine (Life Technologies) supplemented with 10% FCS, 1 mM HEPES, and 100 μg/ml streptomycin/penicillin. They were passaged every 3 to 4 days. Cells were tested

negative for *Mycosplasma* contamination. Cells were not recently authenticated.

## Bacterial strains and culture

NTHi strain RdKW20 (Bishop-Hurley et al, 2005; Domenech et al, 2016) and *Moraxella catarrhalis* strain 25293 (Blakeway et al, 2014) were purchased from the American Type Culture Collection (ATCC). *Staphylococcus aureus* strain 160201753001 and *Pseudomonas aeruginosa* strain 160601067201 from blood culture were provided by Professor Claire Poyart (Cochin Hospital). NTHi, *S. aureus* and *P. aeruginosa* were cultured on chocolate agar plates and *M. catarrhalis* was cultured on brain-heart infusion (BHI) agar plates. Plates were incubated for 24 h at 37 °C until colonies appeared. All strains were grown in LB medium but for NTHi, this was also supplemented with 10 µg/ml hemin and 1 µg/ml nicotinamide adenine dinucleotide (NAD).

## Human rhinovirus production

Human rhinovirus 16 (HRV16) (VR-283, strain 11757, lot 62342987) was purchased from the ATCC and stocks were produced by infecting HeLa Ohio cells in virus medium (DMEM GlutaMax containing 25 mM D-glucose supplemented with 10% FCS and 2 mM L-glutamine) as described previously (Bennett et al, 2012). Briefly, HeLa Ohio cells were grown to 80% confluence at 37 °C and infected with 5 ml HRV16 or control media for 1 h at room temperature with agitation. The remaining solution was made to 10 ml and the cells with HRV16 left for 48 h at 37 °C to allow for 90% cytopathic effect to develop. Supernatants were then clarified by centrifugation and filtration (Bennett et al, 2012) and 1 ml stocks were produced and stored at −80 °C. To UV inactivate HRV16 it was treated with UV light (1000 mJ/cm$^2$) for 20 min. Inactivation was confirmed by adding the inactivated virus to HeLa Ohio cells and checking for cytopathic effect.

## Antibodies and reagents

The following primary antibodies were used: purified rabbit anti-SRBCs (IGN Biochemicals), mouse anti-tubulin alpha (clone DM1A, Sigma, T9026), mouse anti-human EEA1 (BD Transduction Laboratories, 610456), mouse monoclonal CD63 (clone TS63, Eurobio), mouse anti-human LAMP1 (clone H4A3, BD Bioscience) and mouse anti-human ARL5b (clone 2D7, NovusBio). DAPI was from Sigma (D9542). Secondary antibodies were: Alexa Fluor 488, Cy3/5-labeled F(ab')$_2$ anti-mouse or rabbit IgG; horseradish peroxidase (HRP)-labeled anti-mouse and anti-rabbit IgG (Jackson Immunoresearch). siRNA sequences were: 5′CGU ACG CGG AAU ACU UCG A3′ (siLuciferase), 5′GUU CAU CAU UCU UGU UGU U3' (siARL5b.1) and 5′CUC AUG AGG AUU UAC GGA A3′ (siARL5b.2). Plasmid expressing a wild-type or T30N mutated ARL5b were obtained from Paul Gleeson's laboratory (University of Melbourne, Australia).

## Quantification of the TCID$_{50}$

HeLa Ohio cells were cultivated in 96-well plates at $1 \times 10^5$ cells/well for 24 h. HRV16 was diluted tenfold from undiluted to $10^{-9}$ in virus medium and 50 µl of each dilution was added to the cells in eight replicate wells. 50 µl of virus medium was added to two rows of control wells in eight replicate wells per group. Cultures were incubated for 4 days at 37 °C until cytopathic effect was observed in 50% of wells. TCID$_{50}$ was calculated using the Spearman–Karber formula as previously outlined (Bennett et al, 2012).

## HRV16 and bacterial infection of macrophages

Macrophages were washed once in PBS and rested in virus medium. HRV16, HRV16$^{UV}$ or mock-infected (MI) supernatants were added to the macrophages and placed at room temperature for 1 h with agitation to achieve $1 \times 10^7$ TCID$_{50}$/ml. Cultures were then washed with virus medium and rested in macrophage medium overnight.

NTHi, *M. catarrhalis, S. aureus* or *P. aeruginosa* were grown until mid-log growth phase, centrifuged at $1692 \times g$ for 5 min and re-suspended in 1 ml phagocytosis medium (RPMI 1640 supplemented with 2 mM L-glutamine). Bacteria were added to macrophages pre-treated with HRV16, HRV16$^{UV}$ or MI to achieve a MOI of 10/cell or 40/cell (HRV16 only). A higher MOI was used in HRV16-treated cells to compensate for the phagocytic defect (Jubrail et al, 2020) and allow a similar number of bacteria to be uptaken. Cultures were then centrifuged at $602 \times g$ for 2 min and placed at 37 °C, 5% CO$_2$ for 1 h. Cultures were then washed with PBS and treated with 100 µg/ml gentamicin (NTHi, *S. aureus*, *P. aeruginosa*) or 20 µg/ml (*M. catarrhalis*) for 20 min. The 1 h cultures were washed and lysed in saponin as previously described (Jubrail et al, 2016) and colony-forming units (CFU) estimated using the Miles-Misra technique (Miles et al, 1938). The remaining cultures were left in 2 µg/ml gentamicin for 0.5 h, 3.5 h or 24 h and treated in the same manner to determine intracellular CFU.

## HRV16 infection of HeLa Ohio

HeLa Ohio were washed once and rested in virus medium (HeLa Ohio medium without streptomycin/penicillin). HRV16, or MI supernatants, were added to the cells and placed at room temperature for 1 h with agitation to achieve $1 \times 10^5$ TCID$_{50}$/ml. Cultures were then washed with virus medium and rested in virus medium for 8 h or the indicated amount of time.

## Quantification of HRV16-induced cell death

HeLa Ohio cells were cultivated in 96-well plates at $1 \times 10^5$ cells/well for 24 h. Supernatants from HeLa Ohio infected or not with HRV16, overexpressing or not a WT or a mutant ARL5b, or knocked-down or not for ARL5b were diluted ½ in virus medium, and 50 µl of each dilution were added to the cells in four to eight replicate wells. Cultures were incubated for 2 days at 37 °C until a cytopathic effect was observed. Data are presented as the number of wells with visible cytopathic effect (dead) or no visible cytopathic effect (Live).

## Measurement of phagolysosome activity

Macrophages were infected with HRV16 or controls as described above. After overnight rest cultures were washed with PBS and challenged with either DQ-BSA or H$_2$DCFDA-Oxyburst IgG-opsonized carboxylate beads (kind gift from Dr. David Russell, Cornell University, USA) for up to 2 h (Dumas et al, 2015;

Podinovskaia et al, 2013). Shortly, DQ-BSA is a derivative of BSA labeled with a self-quenched fluorescent dye which, upon hydrolyzation in the phagosome, emits a fluorescent signal. H2DCFDA-Oxyburst compounds are non-fluorescent until they are oxidized to the corresponding fluorescein derivatives, allowing the detection of the presence of ROS. At each time point, cultures were washed with phagocytosis medium and fixed in 4% paraformaldehyde (Sigma-Aldrich) on ice for 45 min. They were then treated with 0.05 M $NH_4Cl/PBS1\times$ for 7 min and detached. Analysis was performed using the BD Fortessa through the APC (calibrator) and Alexa Fluor 488 (sensor) channels, acquiring 10,000 events per sample.

## Phagocytosis and phagosome staining

Macrophages were challenged with IgG-opsonized SRBC for up to 60 min. SRBCs were washed in PBS/BSA 0.1% and opsonized for 30 min with rotation in rabbit-IgG anti-SRBCs. They were further washed, re-suspended in phagocytosis medium and added to macrophages to give approximately ten SRBCs per cell. The plates were centrifuged at room temperature at $502 \times g$ for 2 min and then placed at 37 °C for various time points. At each time point, cells were washed with room temperature phagocytosis medium and fixed in 4% paraformaldehyde (PFA) at room temperature for 15 min and then treated with 0.05 M $NH_4Cl/PBS1\times$ for 10 min.

## Microscopy

Cultures were washed in PBS1×/2% FCS and external SRBCs were labeled for 30 min with F(ab')$_2$ anti-rabbit IgG Alexa Fluor 488 in PBS1× /2% FCS. Cells were then washed with PBS1× /2%FCS and re-fixed in 4% paraformaldehyde (PFA) for 15 min at room temperature and then treated with 0.05 M $NH_4Cl/PBS1\times$ for 10 min before being permeabilized in PBS1×/2%FCS/0.05% saponin (permeabilization buffer). Recruitment of markers around phagosomes was then detected using either anti-human EEA1 (BD), anti-CD63 (Eurobio) or anti-human LAMP1 (BD) in permeabilization buffer for 45 min. After washing, cultures were stained with Cy3-labeled F(ab')$_2$ anti-mouse IgG (to detect markers) or Cy5-labeled F(ab')$_2$ anti-rabbit IgG (to detect intracellular SRBCs) in the same buffer for 30 min. After washing in permeabilization buffer cells were stained with DAPI for 5 min and mounted using Fluormount G (Interchim). To quantify phagocytosis, the numbers of internalized and bound SRBCs were counted in 30 cells randomly chosen on the coverslips. The phagocytic index, i.e., the mean number of internalized SRBCs per cell, was calculated. The index obtained for virus-treated cells was expressed as a percentage of the index obtained for control cells. The index of association, which corresponds to the number of bound and internalized SRBCs per cell, was also calculated. To determine the number of internalized SRBCs with specific recruitment of EEA1, CD63 or LAMP1, the internalized SRBCs were counted and scored as positive or negative for each marker. The percentage of positive phagosomes was then calculated. Image acquisition was performed on an inverted wide-field DMI6000 microscope (Leica Microsystems, Wetzlar) with a 100× (1.4 NA) objective and an Orca Flash 4LT+ camera (Hamamatsu Photonics). Z series of images were taken at 0.3 µm z-step increments.

To analyze EEA1 or CD63 staining after HRV16 challenge and overnight rest, cultures were washed in PBS1×/2% FCS/0.05%

saponin (permeabilization buffer) and labeled for 45 min with anti-EEA1 or anti-CD63 in the same buffer. Cells were then washed with permeabilization buffer and incubated with Cy3-labeled F(ab')$_2$ anti-mouse IgG in the same buffer for 30 min. After washing, cells were stained with DAPI for 5 min and mounted using Fluormount G (Interchim). Images were acquired with the conditions described above.

## Conventional electron microscopy

Cells seeded on glass coverslips were chemically fixed 24 h in 2.5% (v/v) glutaraldehyde in 0.1 M cacodylate buffer (pH 7.4), post-fixed 45 min in the dark (4 °C) with 1% (w/v) osmium tetroxide supplemented with 1.5% (w/v) potassium ferrocyanide, dehydrated in ethanol and embedded in Epon as described in (Hurbain et al, 2017). Ultrathin sections of 60–70 nm thickness were prepared with a Reichert UltracutS ultramicrotome (Leica Microsystems), post-stained with 4% aqueous uranyl acetate (10 min, RT) and lead citrate (1 min, RT). Electron micrographs were acquired using a Tecnai Spirit G2 transmission electron microscope (FEI, Eindhoven, The Netherlands) operated at 80 kV and equipped with a 4k CCD camera (Quemesa, Olympus, Münster, Germany).

## Quantification of staining in macrophages

To quantify the intensity of staining of EEA1 and CD63 in macrophages and the total number of vesicles in macrophages, a macro was developed that quantified the two parameters on a per cell basis. Quantification was performed using ImageJ software (Rasband, 1997–2018). In order to properly assess the number and intensity of single endosomes, Z stack images of the entire thickness of cells were acquired. On a duplicate stack, after z-max projection, the number of cells to be quantified in the field of view was manually selected (all cells completely within the field were included). For each selected cell, a freehand selection was made to fix the cell boundaries as well as the center of the nucleus, and after a denoising process, a top-hat filter was applied, and a mask was created by automatic threshold (Otsu algorithm). The number of endosomes and their respective intensity were then calculated from the original image. The results were reported as a summary for each cell in ImageJ, and plotted using software Graphpad prism®.

## Western blots

Macrophages were lysed with lysis buffer (20 mM Tris HCl, pH 7.5, 150 mM NaCl, 0.5% NP-40, 50 mM NaF and 1 mM sodium orthovanadate supplemented with complete protease inhibitor cocktail (Roche Diagnostic)) for 15 min. Lysates were centrifuged at $16,100 \times g$ for 10 min at 4 °C. The supernatants were collected and stored at −20 °C and an equal amount of protein (BCA dosage kit, Pierce) was analyzed by SDS-PAGE. Proteins were transferred onto a polyvinylidene difluoride (PVDF) membrane (Millipore) at 4 °C for 100 min at 100 V and incubated in blocking solution (0.1% Tween-20 supplemented with 5% milk or BSA in TBS1×) for 2 h. Blots were rinsed with TBS/0.1% Tween-20, and primary antibodies were incubated in the blocking solution overnight or for 2 h as required. The membrane was further washed and incubated with HRP-coupled secondary antibodies in blocking buffer for 45 min. Detection was performed using ECL

Dura Substrate (GE Healthcare) and bands imaged by Fusion (Vilber Lourmat) and quantified in ImageJ.

## CD63 surface staining by flow cytometry

Macrophages were infected with HRV16 or MI control as described above. After overnight rest, cultures were washed with PBS and stained with mouse monoclonal anti-CD63 on ice for 45 min in PBS1 × X/2% FCS. They were washed with PBS1×/2% FCS and stained with Alexa Fluor 488-labeled F(ab')$_2$ anti-mouse IgG for 30 min in PBS1×/2% FCS on ice. They were then fixed in 4% paraformaldehyde on ice for 45 min, treated with 0.05 M NH$_4$Cl/PBS1× for 10 min, and then analyzed by BD Fortessa using the Alexa Fluor 488 channel acquiring 10,000 events per sample.

## Next-generation RNA sequencing

Human monocyte-derived macrophages (hMDMs) were challenged with HRV16, HRV16[UV] or mock-infected and after overnight rest were lysed for total RNA using Trizol (Sigma). Lysed samples were then stored at −80 °C until processing. RNA was extracted using the Qiagen RNeasy Mini Kit (Qiagen) according to the manufacturer's instructions.

RNA integrity was analyzed on the Fragment Analyzer platform (AATI, IA, USA) using standard sensitivity RNA kit. RNA was then diluted to 20 ng/µL and used as input to create mRNA libraries using TruSeq Stranded mRNA kit (Illumina, CA, USA) with dual indexing following standard instructions. Libraries were validated on the Fragment Analyzer platform (AATI, IA, USA) using standard sensitivity NGS fragment analysis kit, and the concentration was determined using Quant-iT dsDNA High Sensitivity assay kit on the Qubit fluorometer (ThermoFisher, MA, USA). Sample libraries were pooled in equimolar concentrations and diluted and denatured according to Illumina guidelines. Sequencing was performed using High Output 2 × 76 bp kit on an Illumina NextSeq500.

RNA sequencing fastq files were processed using bcbio-nextgen (version 0.9.9) where reads were mapped to the human genome build hg38 (GRCh38.78) using hisat2 (version 2.0.4) yielding between 17.3–38.3 million mapped reads (average 25.5 million) with a mapping frequency ranging between 86–95% (average 90%) per sample. Sequence quality was evaluated by inspection of phred scores, per N base content, per sequence GC content, duplication levels, genomic distribution of mapped reads, and gene coverage. Gene level quantifications, counts and transcript per million (TPM), were generated with featurecounts (version 1.4.4) and sailfish version 0.10.1), respectively, all within bcbio. Differential gene expression was assessed in R (version 3.3.1) with DESeq2, using raw counts as input. Genes were considered significantly differentially expressed if they had a FDR < 0.05 using a Benjamini-Hochberg method for multiple testing correction. ArrayStudio version 10 (OmicSoft, Cary, NC) was used for further data analysis. Data were analyzed through the use of IPA (QIAGEN Inc., https://digitalinsights.qiagen.com/products-overview/discovery-insights-portfolio/analysis-and-visualization/qiagen-ipa/). Results obtained are presented in three different files showing: comparison of HRV16 VS MI (Dataset EV1), comparison of HRV16-UV VS MI (Dataset EV2), and the 160 genes showing a larger variation than the normal spread (Dataset EV3).

## qPCR

Macrophages or HeLa Ohio were infected with HRV16 or MI control as described above. After overnight rest for macrophages or 8 h rest for HeLa Ohio, cultures were washed with PBS and RNA was extracted as previously described (Chomczynski and Sacchi, 1987). Briefly, hMDMs were washed with PBS at room temperature and lysed using Trizol reagent (ThermoFisher Scientific). Proteins (organic phase) and RNA and DNA (aqueous phase) were separated using chloroform for 2 min at room temperature followed by 15 min centrifugation at 4 °C at 12,000 × g. The aqueous phase was collected and isopropanol was added to precipitate RNA and incubated for 10 min at room temperature. Samples were centrifuged for 20 min at 4 °C at 15,000 × g and the pellet of RNA was washed with 75% ethanol and centrifuged for a further 5 min at 4 °C at 10,000 × g. The pellet was dried at room temperature and re-suspended in pure water and warmed at 55 °C for 5 min. HeLa Ohio were infected with HRV16 or MI control as described above. At the indicated time post infection, cultures were washed with PBS and RNA was extracted using a kit and following the manufacturer's instruction (Qiagen). The total amount of RNA was quantified using nanodrop. For reverse transcription, 1 µg of mRNA was retro-transcribed into DNA using SuperScript II Reverse Transcriptase (ThermoFisher Scientific). qPCR was performed using the LightCycler 480 SYBR Green I Master (Roche) with specific oligos to detect ARL5b (F: ATGGGGCTGATCTTCGCCAAACT; R: CACCAATCCGGGAGGTCATCCACTCT) or HRV (F: GTGAAGAGCCGCGTGTGCT; R: GCTGCAGGTTTAAGGTTAGCC) with 18 S RNA (F: AGGAATTGACGGAAGGGCAC; R: GGACATCTAAGGGCATCACA) or TBP (F: GAGCCAAGAGTGAAGAACAGTC; R: GCTCCCCACCATATTCTGAATCT) as control.

## siRNA treatment

Macrophages at day 7 or HeLa Ohio at 24 h post seeding were washed twice with the corresponding culture medium and kept in new medium at 37 °C. The siRNA solution was prepared in OptiMEM medium (GlutaMAX supplemented, Gibco), containing lipofectamine RNAiMAX reagent (Invitrogen) and siRNA at a final concentration of 240 nM. siRNA was added to each well and cultures left for 24 h at 37 °C before being infected with HRV16 or MI control and processed for further analysis.

## Plasmid treatment

HeLa Ohio at 24 h post seeding were washed with corresponding culture medium and kept in medium at 37 °C. The plasmid solution was prepared in OptiMEM medium (GlutaMAX supplemented, Gibco), containing lipofectamine 2000 reagent (Invitrogen) and 0.8 µg of plasmids per well. Plasmids were added to each well and cultures left for 24 h at 37 °C before being infected with HRV16 or MI control and processed for further analysis. The plasmids used encoded either a WT ARL5b fused to mCherry (pARL5b WT) or an inactive ARL5b fused to GFP (pARL5b T30N). The respective empty control plasmid was used (pCtrl WT or pCtrl T30N, respectively).

## Quantification and statistical analysis

Statistical tests were performed using GraphPad prism® software unless indicated otherwise in the figure legend. All statistical tests,

number of independent experiments performed, and dispersion measures (SD or SEM) are listed in the figure legends and significance is indicated as follows: $*P < 0.05$; $**P < 0.01$; $***P < 0.001$.

## Data availability

Source data are available with this manuscript for data tables and on BioStudies for the images shown in the manuscript under the accession number S-BIAD931. RNA sequencing raw data underlying the findings described in this manuscript may be obtained in accordance with AstraZeneca's data-sharing policy described at https://astrazenecagrouptrials.pharmacm.com/ST/Submission/Disclosure.
Any researcher might ask to have access to the data by using the "Enquiries about Vivli Member Studies" (https://vivli.org/members/enquiries-about-studies-not-listed-on-the-vivli-platform/) form and including the publication title and data accession number GSF1284459. Selected data of interest are also presented in Dataset EV1 and EV2.

## Peer review information

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

## Acknowledgements

We acknowledge the contribution of Ana Filipa Azevedo Campos (ERASMUS+ programme, University of Porto) and Prof. Juliana Felix de Melo (University of Piaui, Bresil) for their contribution to some set-up experiments. We thank Dr. Celine Plainvert and Prof. Claire Poyart for providing bacterial strains, Prof. David Russell for providing the DQ-BSA and $H_2$DCFDA-Oxyburst beads, Dr. Lisa Parker for providing tables to calculate HRV16 $TCID_{50}$ and Prof. Paul Gleeson for the ARL5b plasmids. We thank the CYBIO facility and the IMAG'IC facility of Institut Cochin that is part of the national France-BioImaging infrastructure supported by the French National Research Agency (ANR-10-INBS-04). Work in the laboratory of FN was supported by grants from CNRS, INSERM, Université Paris Descartes (now Université Paris Cité), and a collaborative grant with AstraZeneca/Inserm Transfert. SFD work was supported by a grant from FRM (ARF202110013926) and a Marie Skłodowska-Curie Actions postdoctoral fellowship (Project MacroRhino). CD work was supported by the French National Research Agency through the "Investments for the Future" program (France-BioImaging, ANR-10-INBS-04) and we acknowledge the PICT-IBiSA, member of the France-BioImaging infrastructure, supported by the French National Research Agency (ANR-10-INBS-04) and supported by the CelTisPhyBio Labex (N° ANR-11-LABX-0038) part of the IDEX PSL (N°ANR-10-IDEX-0001-02 PSL).

## Author contributions

**Suzanne Faure-Dupuy**: Conceptualization; Data curation; Formal analysis; Funding acquisition; Validation; Investigation; Visualization; Methodology; Writing—original draft; Writing—review and editing. **Jamil Jubrail**: Conceptualization; Data curation; Formal analysis; Validation; Investigation; Visualization; Methodology; Writing—original draft. **Manon Depierre**: Data curation; Formal analysis; Investigation; Visualization. **Kshanti Africano-Gomez**: Data curation; Formal analysis; Investigation; Visualization. **Lisa Öberg**: Conceptualization; Resources; Data curation; Formal analysis; Validation; Visualization; Methodology; Writing—review and editing. **Elisabeth Israelsson**: Data curation; Formal analysis; Investigation; Visualization; Writing—review and editing. **Kristofer Thörn**: Data curation; Formal analysis; Investigation; Visualization; Writing—review and editing. **Cédric Delevoye**: Resources; Data curation; Formal analysis; Visualization; Methodology; Writing—review and editing. **Flavia Castellano**: Data curation; Formal analysis; Investigation; Visualization. **Floriane Herit**: Data curation; Formal analysis; Investigation; Visualization. **Thomas Guilbert**: Formal analysis; Methodology. **David G Russell**: Resources; Methodology. **Gaell Mayer**: Data curation; Formal analysis; Writing—review and editing. **Danen M Cunoosamy**: Data curation; Formal analysis. **Nisha Kurian**: Conceptualization; Formal analysis; Methodology; Writing—review and editing. **Florence Niedergang**: Conceptualization; Resources; Data curation; Supervision; Funding acquisition; Investigation; Visualization; Methodology; Writing—original draft; Project administration; Writing—review and editing.

## Disclosure and competing interests statement

LO, EI, KT, GM, DC, and NK work or worked for AstraZeneca which develops treatments and vaccines against pathogens. The remaining authors declare no competing interests.

# Expanded View Figures

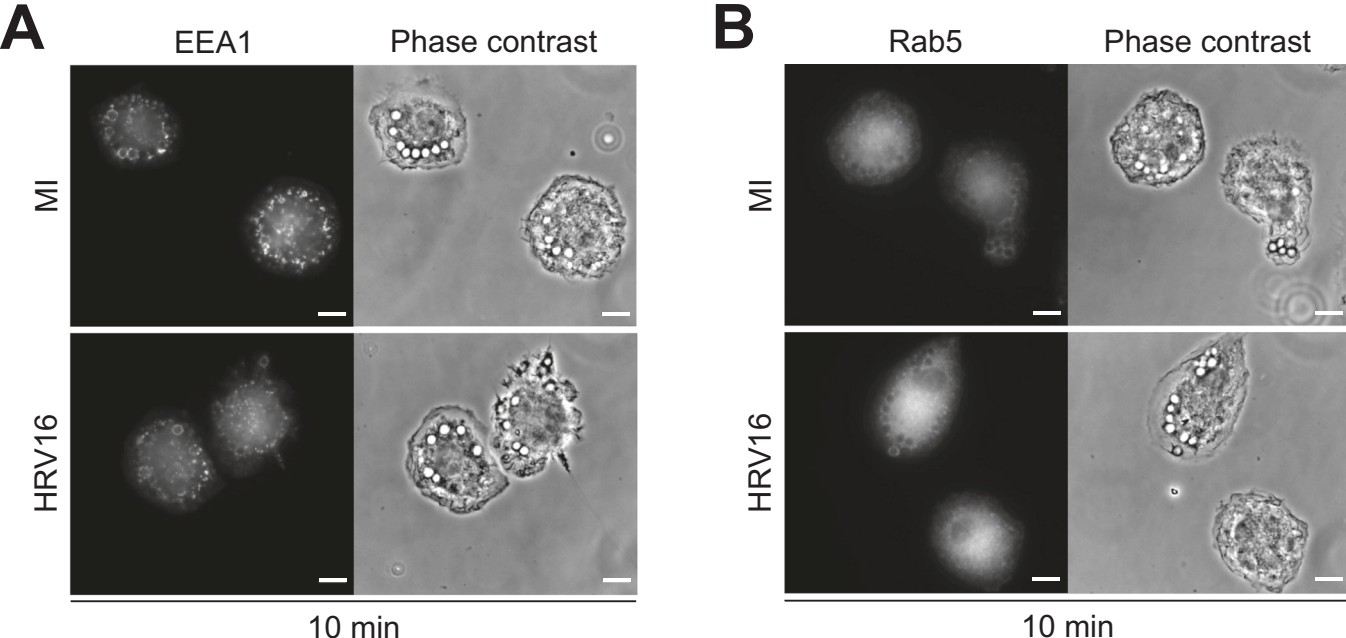

**Figure EV1. HRV16 does not impair EEA1 or Rab5 recruitment to phagosomes after 10 min.**

(A, B) hMDMs were challenged with HRV16 or MI and then exposed to IgG-opsonized sheep red blood cells for 10 min and either stained for (A) EEA1 or (B) Rab5. Representative images of (A) EEA1 or (B) Rab5 staining for MI (upper row) and HRV16-treated cells (lower row) are shown. Data information: (A, B) Scale bar represents 10 μm.

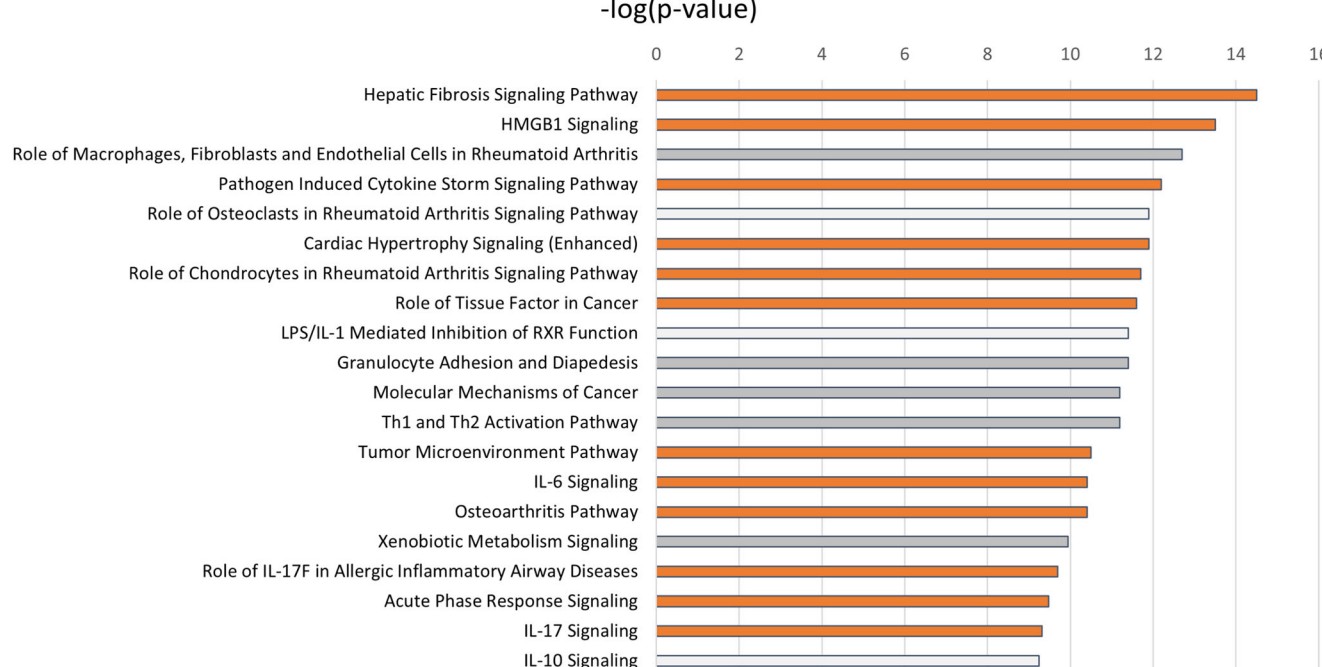

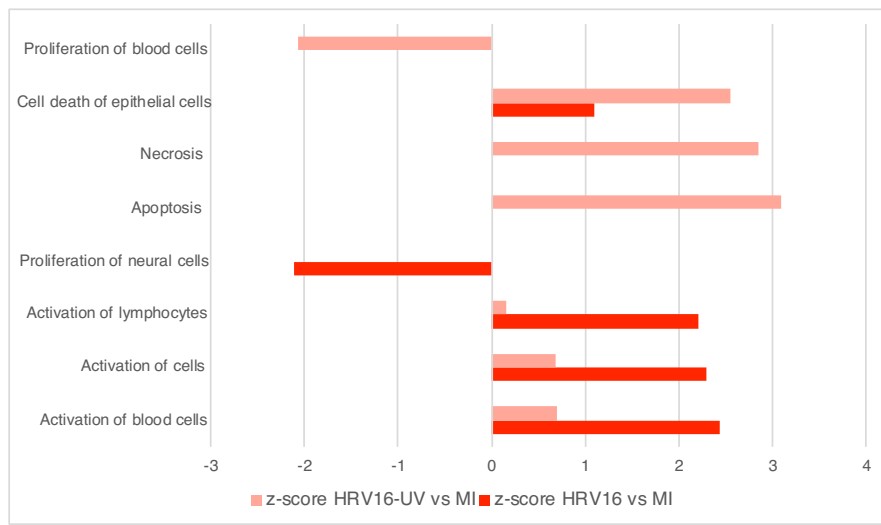

**Figure EV2. Transcriptomic analysis of HRV16-treated macrophages.**

hMDMs were challenged with HRV16, HRV16$^{UV}$ or MI and characterized by RNA sequencing, $n = 6$. Data were analyzed using Ingenuity Pathway Analysis. (**A**) Most dysregulated pathways in HRV16-treated cells compared to MI. Orange bars indicate a predicted significantly activated pathway with a z-score >2. White bars indicate a predicted non-significantly activated pathway with a $-2$>z-score >2. Gray bars indicate that the pathway does not have a directionality that allow activation/inhibition to be predicted. (**B**) Dysregulated pathways in the HRV16-treated or HRV16$^{UV}$-treated cells compared to MI as determined from the 160 genes showing a larger variation than the normal spread when comparing the fold changes induced by HRV16 or HRV16$^{UV}$. Data information: $P$ values and z-scores were evaluated with Ingenuity Pathway Analysis. Right-tailed Fisher's exact test statistical analysis was performed.

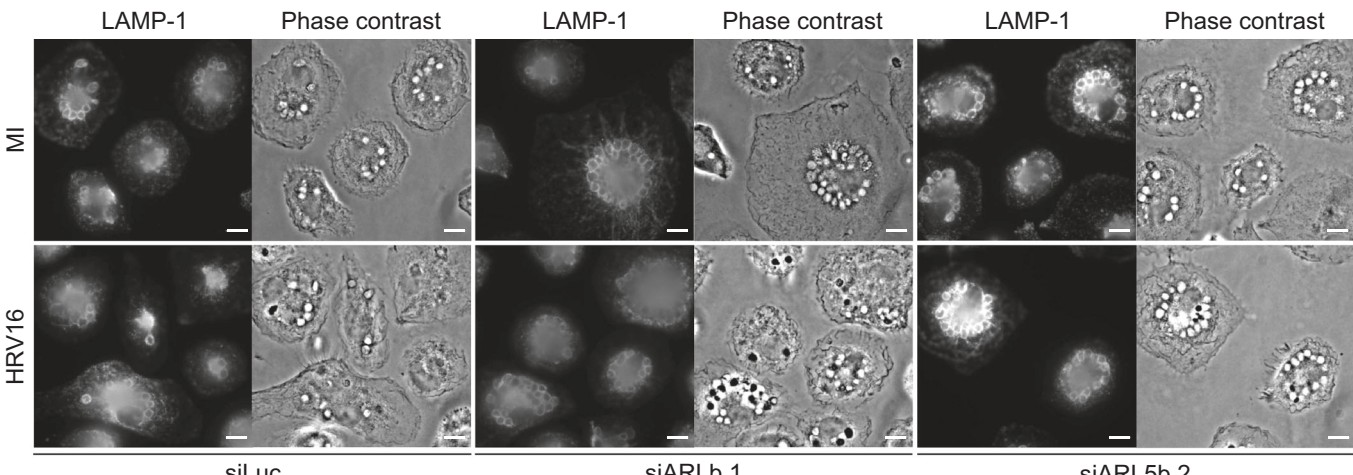

**Figure EV3. ARL5b depletion restores the recruitment of LAMP-1 to phagosomes in HRV16-treated macrophages.**

hMDMs were transfected with siRNA against luciferase (siLuc, control) or ARL5b (siARL5b.1 and siARL5b.2) and challenged with HRV16 or MI. Cells were exposed to IgG-opsonized sheep red blood cells for 60 min and LAMP-1 was stained. Representative images of LAMP-1 staining for MI (upper row) and HRV16-treated cells (lower row) for each siRNA are shown. Data information: Scale bar represents 10 μm.

