## [Peer Review File · EMBO Reports]

ARL5b inhibits human rhinovirus 16 propagation and impairs macrophage-mediated bacterial clearance

Suzanne Faure-Dupuy, Jamil Jubrail, Manon Depierre, Kshanti Africano-Gomez, Lisa Öberg, Elisabeth Israelsson, Kristofer Thörn, Cedric Delevoye, Flavia Castellano, Floriane Herit, Thomas Guilbert, David Russell, Gaëll Mayer, Danen Cunoosamy, Nisha Kurian, and Florence Niedergang

Corresponding author(s): Florence Niedergang (florence.niedergang@inserm.fr)

Review Timeline:

Submission Date:	1st May 23
Editorial Decision:	8th May 23
Appeal Received:	12th May 23
Editorial Decision:	19th May 23
Editorial Decision:	4th Jul 23
Revision Received:	14th Nov 23
Editorial Decision:	15th Dec 23
Revision Received:	21st Dec 23
Accepted:	3rd Jan 24

Transaction Report:

Dear Dr. Niedergang,

Thank you for the submission of your manuscript to EMBO reports. I have now read and discussed your work with my colleagues here, and I regret to say that we all agree that it is not well suited for our journal.

We appreciate that your study reports that HRV16 infection increases the expression of Arl5b/Arl8 in macrophages, which limits the maturation of phagolysosomes and promotes bacterial survival. You further show that HRV16 reduces Arl5b expression in epithelial cells, which increases viral secretion and spread.

We acknowledge that your findings indicate that Arl5b restricts viral propagation in epithelial cells and phagolysosome maturation in macrophages. Clearly, your results will be of interest to researchers working closer to the field. However, we also note that Arl5b has been linked to endosome-lysosome and phagosome-lysosome fusion before and its upregulation has also been shown to reduce macrophage-mediated killing of bacteria. We further note that the current data do not address how HRV16 alters the expression of Arl5b upon infection. Taking these considerations into account, we overall feel that the manuscript does not provide the type of compelling advance that EMBO reports seeks to publish. We have therefore decided not to proceed with in-depth peer review.

In the interest of your manuscript and your time, I am providing you with an editorial decision on your manuscript that will allow you to submit it elsewhere without further delay. I am sorry to disappoint you on this occasion, and I thank you once more for your interest in our journal.

Yours sincerely,

** As a service to authors, EMBO Press provides authors with the ability to transfer a manuscript that one journal cannot offer to publish to another journal, without the author having to upload the manuscript data again. To transfer your manuscript to another EMBO Press journal using this service, please click on Link Not Available

Dear Dr Rembold,

I thank you for your message and your feedback.

I just would like to point out that ARL5B has not been related to endo-lysosome fusion nor phagosome-lysosome fusion before.

These are functions that were described for ARL8.

The literature is indeed confusing, but the Arl8a and Arl8b proteins are not the same as Arl5b (see Rosa Ferreira and Munro Dev Cell 2011, PMC3240744).

The Arl5b proteins are described as Golgi localised (which we also confirmed - results not included in our manuscript) and involved in retrograde transport in (Houghton et al, Exp Cell research 2012, 10.1016/j.yexcr.2011.12.023).

Therefore all the articles describing the link with lysosomes from the Munro, Bonifacino, Botelho teams and others, cannot support what we observed in our study.

We believe that our work brings the novelty of the role of Arl5b in bacterial clearance and phagolysosome maturation.

I thank you for your attention,

Yours sincerely,

Florence Niedergang

Florence Niedergang, PhD
Institut Cochin
Team Biology of Phagocytes, Infection and Immunity
Batiment Méchain, 3rd floor
Tel. +33(0)1 40 51 64 21
florence.niedergang@inserm.fr <>

Dear Dr. Niedergang,

I apologize for my delayed response but I have now re-read your manuscript and my notes and I have carefully considered your arguments and clarification on Arl5b and its reported functions.

Taking all information into account, I have no objection to seek the input from three referees in the field. I am sure you appreciate, however, that we cannot predict the outcome of the reviewing process, which may turn out to be the same.

I will convert your manuscript into an official Appeal in our manuscript submission system, so that it is 'live' again and can be reviewed.

Please let me know if you agree with this.

Kind regards,
Martina

Martina Rembold, PhD
Editor | EMBO reports
m.rembold@emboreports.org

Dear Florence,

Thank you for the submission of your research manuscript to our journal. As you know, we have now received the full set of referee reports that is copied below.

As you will see, the referees acknowledge that the findings are potentially interesting, but they also raise a number of concerns that need to be addressed. It will be important to analyse potential defects in the endomembrane system and trafficking in more detail, but a TEM analysis is not mandatory for publication here, as referee also indicates in his/her report.

Given these constructive comments, we would like to invite you to revise your manuscript with the understanding that the referee concerns (as detailed above and in their reports) must be fully addressed and their suggestions taken on board. Please address all referee concerns in a complete point-by-point response. Acceptance of the manuscript will depend on a positive outcome of a second round of review. It is EMBO reports policy to allow a single round of revision only and acceptance or rejection of the manuscript will therefore depend on the completeness of your responses included in the next, final version of the manuscript.

As discussed via e-mail, please change the text in the Data availability section to clearly state that the data will be made available to any researcher asking for it (via website or personal contact). Please also include supplementary figures and/or tables with all the differentially expressed genes corresponding to Figure 4. I assume that this will result in complex and data-rich tables. Please upload these as 'Dataset EV#' and provide the legend in a separate tab of the .xls file.

We realize that it is difficult to revise to a specific deadline. In the interest of protecting the conceptual advance provided by the work, we recommend a revision within 3 months (October 4th). Please discuss the revision progress ahead of this time with the editor if you require more time to complete the revisions.

I am also happy to discuss the revision further via e-mail or a video call, if you wish.

*******IMPORTANT NOTE:**

We perform an initial quality control of all revised manuscripts before re-review. Your manuscript will FAIL this control and the handling will be DELAYED if the following APPLIES:

- 1) A data availability section providing access to data deposited in public databases is missing. If you have not deposited any data, please add a sentence to the data availability section that explains that.
- 2) Your manuscript contains statistics and error bars based on $n=2$. Please use scatter blots in these cases. No statistics should be calculated if $n=2$.

When submitting your revised manuscript, please carefully review the instructions that follow below. Failure to include requested items will delay the evaluation of your revision.*****

- 1) a .docx formatted version of the manuscript text (including legends for main figures, EV figures and tables). Please make sure that the changes are highlighted to be clearly visible.
- 2) individual production quality figure files as .eps, .tif, .jpg (one file per figure). Please download our Figure Preparation Guidelines (figure preparation pdf) from our Author Guidelines pages <https://www.embopress.org/page/journal/14693178/authorguide> for more info on how to prepare your figures.
- 3) a .docx formatted letter INCLUDING the reviewers' reports and your detailed point-by-point responses to their comments. As part of the EMBO Press transparent editorial process, the point-by-point response is part of the Review Process File (RPF), which will be published alongside your paper.
- 4) a complete author checklist, which you can download from our author guidelines (<<https://www.embopress.org/page/journal/14693178/authorguide>>). Please insert information in the checklist that is also reflected in the manuscript. The completed author checklist will also be part of the RPF.
- 5) Please note that all corresponding authors are required to supply an ORCID ID for their name upon submission of a revised manuscript (<<https://orcid.org/>>). Please find instructions on how to link your ORCID ID to your account in our manuscript tracking system in our Author guidelines

(<<https://www.embopress.org/page/journal/14693178/authorguide#authorshipguidelines>>)

6) We replaced Supplementary Information with Expanded View (EV) Figures and Tables that are collapsible/expandable online. A maximum of 5 EV Figures can be typeset. EV Figures should be cited as 'Figure EV1, Figure EV2' etc... in the text and their respective legends should be included in the main text after the legends of regular figures.

7) At EMBO Press we ask authors to provide source data for the main figures. Our source data coordinator will contact you to discuss which figure panels we would need source data for and will also provide you with helpful tips on how to upload and organize the files.

Additional information on source data and instruction on how to label the files are available <<https://www.embopress.org/page/journal/14693178/authorguide#sourcedata>>.

8) The journal requires a statement specifying whether or not authors have competing interests (defined as all potential or actual interests that could be perceived to influence the presentation or interpretation of an article). In case of competing interests, this must be specified in your disclosure statement. Further information: <https://www.embopress.org/competing-interests>

9) Figure legends and data quantification:

- the name of the statistical test used to generate error bars and P values,
 - the number (n) of independent experiments (please specify technical or biological replicates) underlying each data point,
 - the nature of the bars and error bars (s.d., s.e.m.)
- If the data are obtained from n {less than or equal to} 5, show the individual data points in addition to the SD or SEM.
- If the data are obtained from n {less than or equal to} 2, use scatter blots showing the individual data points.

10) Our journal encourages inclusion of *data citations in the reference list* to directly cite datasets that were re-used and obtained from public databases. Data citations in the article text are distinct from normal bibliographical citations and should directly link to the database records from which the data can be accessed. In the main text, data citations are formatted as follows: "Data ref: Smith et al, 2001" or "Data ref: NCBI Sequence Read Archive PRJNA342805, 2017". In the Reference list, data citations must be labeled with "[DATASET]". A data reference must provide the database name, accession number/identifiers and a resolvable link to the landing page from which the data can be accessed at the end of the reference. Further instructions are available at <<https://www.embopress.org/page/journal/14693178/authorguide#referencesformat>>.

11) As part of the EMBO publication's Transparent Editorial Process, EMBO reports publishes online a Review Process File to accompany accepted manuscripts. This File will be published in conjunction with your paper and will include the referee reports, your point-by-point response and all pertinent correspondence relating to the manuscript.

Kind regards,

Martina

Referee #1:

The manuscript by Jubrail et al. describes impaired phagosome maturation in human monocyte-derived macrophages after rhinovirus infection. They show how infected macrophages fail to degrade phagocytosed bacteria, presumably because of reduced hydrolysis and endosome formation. They report upregulation of ARL5b in the virus-infected macrophages, and knocked down ARL5b can restore bacterial degradation intracellularly. HeLa cells infected with rhinovirus showed increased viral transcripts and decreased ARL5b. Finally, they show that ARL5b inhibits viral propagation but not replication. Overall they provide interesting findings to show dual role of ARL5b in macrophages and epithelial cells. However, the cellular mechanisms governing ARL5b in each cell type remains less clear. Several data figures also need attention:

Major concerns:

1. The group previously (Jubrail et al. 2020) shown that HRV16 reduces phagocytosis in macrophages; however, the results in Fig. 1B to 1E show all 3 groups have the same phagocytose activity at baseline (1 h). The authors should reconcile these findings and their implication in the new study.
2. Fig. 1F: the authors should describe how altering pH leads to fluorescent color changes.
3. The authors measured endosome markers at different time points (from 10 to 60 min) but Fig. 2 only shows presentative figures at 60 min. Other representative figures using different time points should be provided for each of the quantification data points.
4. Fig. 2: Images of Rab5 and Rab7 conversion should be provided.
5. Fig. 3D: WB of the target proteins is unclear.
6. The image quality of MI with CD63 in Fig. 3F is unclear.
7. Though the authors identified ARL5b as a regulator for endosome function, the rationale to target this gene and in-depth pathway and interactome analyses is lacking.
8. The Y-axis label in Fig. 4B and the meaning of the color code index in Fig. 4C are missing.
9. The results in Fig. 4F should show in the same uncropped blot for each target protein and with biological repeats.
10. The results in Fig. 5B should be normalized to HRV16 + siLuc instead of normalizing MI and HRV16 separately.
11. Photos in Fig. 5C are not comparable. MI + siARL5b.2 seems to have the highest intensity among others.
12. EEA1 and LAMP1 should also be examined in Fig. 5 to better understand which steps of endosomal formation are affected by reducing ARL5b.
13. Using only one cell line (HeLa) for experiments in Fig. 6 is insufficient to conclude the role of ARL5b. Other cell lines or primary lung epithelial cells are recommended.
14. General stats: The control conditions presented lack standard deviation, suggesting the experiments were individually normalized. A more rigorous analysis includes averaging all control data to 1 and then compare each control data to the average so the control group can have a standard deviation. All the bar graphs should show individual data points (for example, Fig. 2E and 6) and standard deviation (Fig. 4G, 5B, and 5D). There are missing comparison lines in Fig. 2, 4D, and 5 to guide the readers. The flow of the figure legend is hard to follow, especially in Fig. 3.

Minor concerns:

1. Inconsistent graph style in the manuscript. For example, the thickness of the lines, fonts, size of symbols, and color of the bars.
2. Error bars in Fig 3B, 3C, and 3H should use a color other than black and red so people can see.
3. Other bacteria species used in Fig. 1 should also be applied for experiments in Fig. 5F.

Referee #2:

Jubrail and colleagues present interesting data in support of HRV-mediated suppression of bacterial killing my human macrophages. The experiments are appropriate, the data well presented and conclusions justified. I feel that this is of general

interest to the readership of EMBO Reports and to the scientific community in general. I believe the paper can be published with only a few minor revisions:

1. I think the interpretation of the H2DCFDA data may be erroneous. Once the H2DCFDA is oxidized the signal cannot be reversed (ie. reduced again) in the phagosome. So the reduced signal that is observed with time is likely related to the quenching of fluorescein in the acidic compartment as opposed to ROS kinetics. Nevertheless, this is minor and does not change the major interpretation - it is still safe to say ROS is reduced.
2. What is the significance of measuring surface CD63 by flow cytometry? Is surface CD63 relevant to phagosomal CD63?
3. The individually cropped bands of Figure 4F are a little concerning. Please provide a supplementary figure with the uncropped gels.
4. All in all, a nice paper.

Referee #3:

In this article Jubrail et al address the problem of how human rhinoviral infections enhance susceptibility to bacterial super-infections. The question is relevant from the perspective of rhinoviral biology where many questions still remain open. It also has clinical pertinence since rhinoviral and associated bacterial infections are important precursor for asthma and COPD exacerbations in various age groups.

The authors demonstrate defective bactericidal activity of macrophages upon HRV16 exposure, and identify perturbations in the phagosome maturation pathway. With RNA-seq, they further identify gene clusters with altered expression profiles in infected macrophages and characterise the role of one of the genes identified, ARL5b in greater detail. The data suggests dual function for ARL5b in HRV infection - upregulation upon viral infection perturbs phagosome maturation in macrophages, whereas restricts virion secretion in epithelial cells.

The experimental design and execution are commendable, yielding clear results. However, I fear the strength of some of the conclusions drawn exceeds the available evidence, necessitating additional experiments. Please see my detailed comments below.

Figure 1B: The figure indicates that all infection conditions exhibited a nearly identical bacterial load at the initial time point depicted. This finding is perplexing considering the authors' previous demonstration of significant uptake defects during HRV infection, which varied for different bacterial species. In the methods section, it is mentioned that the MOI was adjusted to be four times higher for the viral-infected cases in order to achieve the same initial bacterial load (an explanation that could be included in the results section to prevent reader confusion). However, it remains unclear how this adjustment resulted in identical initial CFUs for all bacterial cases. Previous reports, for example, have shown a difference of nearly two orders of magnitude in initial CFUs between the control and MC, and nearly an order of magnitude between NTHi and MC.

The hydrolytic activity shown in figure 1G: Taking it longer beyond the peak would help to see if it's a delayed rise of activity or reduction in peak activity. This has bearing on questions related to figure 2 mentioned below.

Figure 2 A and C show images taken at 60 min, by which time the enrichment of the markers on phagosomes is lost in controls. Showing an earlier time point when the marker enrichment is visible in controls will help readers appreciate if the phagosomes and enrichment in controls qualitatively appear similar, to the later time point in HRV treated (the data suggests that possibility).

The data presented in Figure 2 can also be interpreted as indicating a slower maturation of the phagocytic process rather than a complete blockage. In the HRV condition, EEA1 reaches its peak at 15 minutes and tends to decrease by the 30-minute mark, while CD63 appears to increase towards control levels by the 30-minute time point. To gain further insights and distinguish between potential issues with slow kinetics or aberrant marker association levels, it would be beneficial to extend the measurements to additional time points. Observing how these markers resolve towards control levels over time, will lead to better understanding of whether the observed phenomena represent slow kinetics and/or disrupted marker associations (phagosome maturation). With the present data, it is challenging to definitively conclude that phagosome maturation is completely arrested. Furthermore, drawing such a definitive conclusion based solely on the three markers utilized seems somewhat overstated. Including the examination of additional markers that play a crucial role in the dynamics of phagocytosis, such as certain Rab-GTPases known to be essential in phagosome maturation, would contribute to a more accurate delineation of the process and any potential aberrations.

A related question pertains to phagocytosis itself. Do HRV-treated cells exhibit defects in red blood cell (RBC) phagocytosis? Are there differences in phagocytosis indices compared to controls? If such differences exist, it raises the question of whether HRV-treated cells initiate the maturation process with significantly divergent phagocytosis loads. Does it have any bearing in say, apparent slower maturation?

The data in figure 3 suggests more than what has been concluded. It is not merely expression and localisation of markers that has been perturbed. Especially with EEA1, it can be appreciated that the vesicles in HRV challenged cells have different characteristics. They are more heterogeneous in size, and a significant portion of the vesicles are much larger than the control vesicles with dramatically higher fluorescence. This is corroborated by the quantifications, which show that there is higher amount of EEA1 without any increase in total number of vesicles. The late endosomal marker also, less strikingly though, suggests alterations to the endosomal compartment. The late endosomes are surface associated, and the vesicle sizes, look potentially quite different.

While a detailed analysis might be beyond the scope of this paper, a more detailed examination of the endomembrane system using light or electron microscopy would be very beneficial. It would aid in determining whether the cell's defects are solely limited to phagosome maturation or if they indicate a broader disruption in endomembrane system organization, which in turn affects phagocytosis.

Such an analysis would also be relevant to the overall narrative as ARL5B, a Golgi-associated GTP-ase, potentially plays a role in both anterograde and retrograde traffic, thereby impacting endomembrane homeostasis. This becomes particularly intriguing considering previous research has shown that disruption of the endomembrane system and traffic can occur as a result of picornavirus infections, including HRV. Elucidating whether these disruptions contribute to bactericidal defects and subsequently increase susceptibility to bacterial infections would greatly enhance the value of this article.

On a similar note, it will be good to know if the significant clusters identified by RNA-seq suggest any perturbation of the membrane trafficking pathways.

Does ARL5B knockdown have any effect on HRV exposed macrophage phagocytosis? Since the ARL5B overexpression system is already made by the authors, its effect on macrophage behavior will be a good control to see.

Figure 6 - while higher cell death with reinfection is an interesting observation, one would like to see the viral titres compared by a plaque assay in HeLaOH or similar to make a conclusion that "Therefore, depletion of ARL5b in HeLa cells increases HRV16 virions secretion and viral propagation."

Sentence 220-222 is written in a confusing manner with an active role ascribed to siLuciferase - it simply needs to say that control siRNA did not alter the behavior. Same issue with 225-227.

In general figure explanations are somewhat inadequate. The legend and the text together still leave the figures not fully explained. As an example, the arrows in Fig 2A are not explained.

There are inconsistencies in language and usage of abbreviations that need to be taken care of.

Reviewer #1:

The manuscript by Jubrail et al. describes impaired phagosome maturation in human monocyte-derived macrophages after rhinovirus infection. They show how infected macrophages fail to degrade phagocytosed bacteria, presumably because of reduced hydrolysis and endosome formation. They report upregulation of ARL5b in the virus-infected macrophages, and knocked down ARL5b can restore bacterial degradation intracellularly. HeLa cells infected with rhinovirus showed increased viral transcripts and decreased ARL5b. Finally, they show that ARL5b inhibits viral propagation but not replication. Overall they provide interesting findings to show dual role of ARL5b in macrophages and epithelial cells. However, the cellular mechanisms governing ARL5b in each cell type remains less clear. Several data figures also need attention:

We thank Reviewer #1 for their acknowledgement of our work and their constructive comments.

Major concerns:

1. The group previously (Jubrail et al. 2020) shown that HRV16 reduces phagocytosis in macrophages; however, the results in Fig. 1B to 1E show all 3 groups have the same phagocytose activity at baseline (1 h). The authors should reconcile these findings and their implication in the new study.

As mentioned by Reviewer #1, we previously showed that HRV16 impairs phagocytosis, leading to less bacteria uptake in HRV16-exposed macrophages (Jubrail et al. 2020). To compensate for this defect, HRV16-treated macrophages were exposed to a higher number of bacteria (MOI=40/cell), as compared with macrophages treated with HRV16^{UV} or mock infected (MI) (MOI=10/cell). The details of the experiments are indicated in the "Material and Methods" section. To facilitate the comprehension of the results presented in Figures 1B to 1E, we have modified the text and indicated the difference of MOI used in these experiments.

2. Fig. 1F: the authors should describe how altering pH leads to fluorescent color changes.

We thank the reviewer for this comment. As suggested, we have added to the "Material and Methods" section a description of how the probes detect the phagolysosomal environment. The text now reads as follow: "Shortly, DQ-BSA is a derivative of BSA labelled with a self-quenched fluorescent dye which, upon hydrolyzation in the phagosome, emits a fluorescent signal. H2DCFDA-Oxyburst compounds are non-fluorescent until they are oxidised to the corresponding fluorescein derivatives, allowing the detection of the presence of ROS." Therefore, the probes used do not directly detect the changes in pH, but rather the hydrolytic activity, which is more efficient at low pH, and the oxidative activity.

3. The authors measured endosome markers at different time points (from 10 to 60 min) but Fig. 2 only shows presentative figures at 60 min. Other representative figures using different time points should be provided for each of the quantification data points.

We agree with Reviewer #1 that a representative image of the different time points would enable an easier reading of the figure. Therefore, we have added the corresponding images to Figure 2. We have additionally provided later time points, as suggested by Reviewer #3, as well as Rab5 and Rab7 staining as suggested by Reviewers #1 and #3.

4. Fig. 2: Images of Rab5 and Rab7 conversion should be provided.

We thank the Reviewer for this comment. Indeed, Rab5 is a marker of early phagosomes, whereas Rab7 is a marker of late phagosomes, which can complement the results already presented with EEA1 and CD63. For this, Rab5 and Rab7 were stained on HRV16-exposed macrophages that underwent a phagocytosis assay. Results showed no significant difference in Rab5 recruitment to phagosomes between MI and HRV16 conditions, at any of the tested time points (i.e. 10 min, 15 min, 60 min, and 120 min post phagocytosis initiation) (new Figures 2C-D). The recruitment of Rab7 to the phagosomes appeared to be enhanced in the HRV16-treated macrophages after 30 min, although the results are not significantly different (new Figures 2G-H). There was no clear difference in Rab7 recruitment on the phagosomes in HRV16- or MI-treated macrophages at 60 min and 120 min post-phagocytosis initiation. The additional markers of phagosome maturation that were monitored indicate that the stalled phagosomes have intermediate phenotypes with normal presence of Rab5, early recruitment of Rab7, prolonged EEA1, Rab7, CD63 and lower recruitment of LAMP1. There is some recovery with time, as the differences are no more significant, but there is still a tendency for the phagosomes to keep higher CD63 and lower LAMP1 recruitment in macrophages treated with HRV16 as compared with the control cells. Therefore, our observation suggest that the phagosome maturation is impaired and not just delayed.

5. Fig. 3D: WB of the target proteins is unclear.

We apologize for this inconvenience. We are not sure what Reviewer #1 means by unclear. As other comments from Reviewer #1 indicate problems in the figures provided that are not visible for us (i.e. comments 6 and 8), there might have been a problem upon uploading the figures. We hope that this problem is now solved.

Concerning the increase of EEA1 expression, which is visible in the HRV16 lane, it has been quantified in 9 independent experiments. The average fold increase is of 1.74, close to the increase in total fluorescence (1.35). This increase in EEA1 is however a robust characteristic of infected cells.

We have additionally relabelled the image to have a clearer indication of which blot correspond to which protein.

6. The image quality of MI with CD63 in Fig. 3F is unclear.

We apologize for this inconvenience. We suppose that the quality of the figures upon uploading and/or transfer to the reviewer was altered. We hope that this will be solved in the revised version.

7. Though the authors identified ARL5b as a regulator for endosome function, the rationale to target this gene and in-depth pathway and interactome analyses is lacking.

We thank Reviewer #1 for this important comment. We have added to the extended view figure corresponding to the results presented in figure 4 the gene pathway analysis of our samples. These results show that there was no highlight of genes implicated in intracellular trafficking and rather clusters of genes playing a role in signaling and cell activation, broadly speaking (new Figure EV2).

Accordingly, the text was modified, and an extended figure added, to include these data and better indicate the rationale behind ARL5b identification. Namely, 33 targets were significantly upregulated by HRV16 exposure as compared to MI and HRV16^{UV} (see table and

heat map of new Figure 4C). As phagosome maturation is linked to endosomal trafficking and that we observed that the Golgi apparatus was modified by HRV16 challenge (new Figures 3J-L), we took an interest in the targets linked to this function. Only ARL5B, a small GTPase involved in the regulation of transport along the endosome-trans Golgi network, was identified as a potential target. To further explain the rationale for selecting ARL5b, we have added the following text to the manuscript: "One of the most consistently and significantly upregulated genes in the HRV16 condition was the small GTPase ARL5b (Figure 4D). ARL5b was reported to control anterograde and retrograde trafficking from and towards the Golgi apparatus (Houghton et al, 2012). As we observed that the endocytic compartments and the Golgi apparatus were impacted by HRV16 (Figure 3J-L), we hypothesized that ARL5b could be involved in the modifications observed."

8. The Y-axis label in Fig. 4B and the meaning of the color code index in Fig. 4C are missing. We apologize if some legends were missing in the version of the Figure 4 that Reviewer #1 got. In the version of the figure we submitted, the Y-axis label in Figure 4B and the meaning of the colour code index in Figure 4C were present. We hope that this problem will be solved with the revised version of the manuscript.

9. The results in Fig. 4F should show in the same uncropped blot for each target protein and with biological repeats.

We agree with Reviewer #1 that an uncropped blot is preferable in Figure 4F. We had initially cropped the images from a larger experiment including siRNA treatment as in Figure 5A. We have now repeated the experiments to increase the number of biological repeats to 6 and to have both MI and HRV16-treated macrophages side by side on the same membrane. Results are provided in Figures 4F and 4G. We observed an increase of ARL5b expression by western blot upon HRV16 exposure in human primary macrophages. Quantification of the different experiments showed a significant induction of ARL5b at the protein level by HRV16 in macrophages.

10. The results in Fig. 5B should be normalized to HRV16 + siLuc instead of normalizing MI and HRV16 separately.

As suggested by Reviewer #1, we have modified the normalization of the WB analysis. Data in figure 5B are now presented as ARL5b intensity normalized to HRV16 + siLuc. Additionally, we have repeated this experiment several times on new primary macrophages derived from various donors and have included the results in figure 5B.

11. Photos in Fig. 5C are not comparable. MI + siARL5b.2 seems to have the highest intensity among others.

We thank the Reviewer for this comment. We have selected new images and made sure they had the same treatment during the image processing. They are included in the new Figure 5C.

12. EEA1 and LAMP1 should also be examined in Fig. 5 to better understand which steps of endosomal formation are affected by reducing ARL5b.

We thank Reviewer #1 for this suggestion. To confirm the effect of ARL5b depletion on phagosome maturation impairment, we performed new experiments. Primary human macrophages were treated for 24h with siRNA control (siLuc) or against ARL5b (siARL5b.1

and siARL5b.2) and then challenged with HRV16 or MI. After an overnight rest, a phagocytosis assay was performed and after 60 min cells were stained for LAMP1. Results showed that in the siLuc condition, HRV16 impairs the recruitment of LAMP1 to the phagosomes (new Figures 5G and EV3), as previously shown in Figures 2I and 2J. Interestingly, when hMDMs were depleted for ARL5b, this phenotype was rescued, and no difference was observed between the MI and HRV16 conditions.

Altogether, we observed that, upon HRV16-challenge, ARL5b depletion rescued the increase of EEA1 intensity, the increased surface localisation of CD63, the impaired bacteria elimination, and the decreased recruitment of LAMP1 to phagosomes. These results confirmed that ARL5b is the key player of HRV16-mediated impairment of the expression and localization of endosomal compartments as well as phagosome maturation.

13. Using only one cell line (HeLa) for experiments in Fig. 6 is insufficient to conclude the role of ARL5b. Other cell lines or primary lung epithelial cells are recommended.

We agree with Referee #1 that different lung epithelial cell models are needed to conclude on the role of ARL5b in epithelial cells. To address this comment, we repeated key experiments of Figure 6 in the BEAS-2B cell line. BEAS-2B are immortalized but non-tumorigenic lung epithelial cells. We performed a kinetic of infection after MI or HRV16 exposure and showed that HRV RNA decreased through time post HRV16 infection (new Figure 6J). These results indicate that HRV16 did not replicate in these cells. Interestingly, at 120 h post-infection, ARL5b expression was increased (new Figure 6K), which is the opposite of what is observed in HeLa Ohio in which the virus actively replicates.

Therefore, the regulation of ARL5b is not identical in all epithelial cells and seems to be correlated with the permissiveness of the cells to viral replication.

We have modified the text accordingly, our conclusions and the discussion to fit better to the results and to remain cautious on generalizing the effect of ARL5b in epithelial cells.

14. General stats: The control conditions presented lack standard deviation, suggesting the experiments were individually normalized. A more rigorous analysis includes averaging all control data to 1 and then compare each control data to the average so the control group can have a standard deviation. All the bar graphs should show individual data points (for example, Fig. 2E and 6) and standard deviation (Fig. 4G, 5B, and 5D). There are missing comparison lines in Fig. 2, 4D, and 5 to guide the readers. The flow of the figure legend is hard to follow, especially in Fig. 3.

As suggested by Reviewer #1, we have modified the normalization process in several panels to show standard deviation in the control conditions, which are normalized to 1 (i.e. Figures 3I, 4E, and 5E). We did not do so in Figures 3E, 4G and 5B as these figures are the quantification of the intensity of the detected band in a Western Blot analysis, and this intensity may vary from the exposure time used. Therefore, it is not possible to compare the intensity between experiments and it is subsequently not possible to normalize on the average of the intensity of the controls in the different experiment.

We have modified bar graphs in Figures 2B, 2D, 2F, 3I, 4E, 5E, and 6 to show individual data points. Standard deviations have been added to Figures 4G and 5B. In the figure 5D, the standard deviations were not visible due to the number of points present. We therefore changed their colours to make them more visible for the readers. Moreover, the comparison lines have been added in all the panels in which they were missing.

Finally, the legends of the figures have been modified for sake of clarity, especially for Figure 3.

Minor concerns:

1. Inconsistent graph style in the manuscript. For example, the thickness of the lines, fonts, size of symbols, and color of the bars.

We have addressed this issue and have normalized all the lines thickness, fonts, size of symbols and colour of the bars in the new version of the figures.

2. Error bars in Fig 3B, 3C, and 3H should use a color other than black and red so people can see.

As suggested by Reviewer #1, we have changed the colours of the error bars throughout the manuscript to (i) make them more visible in Figures 3B, 3C, and, and (ii) to maintain a cohesion in figure representation.

3. Other bacteria species used in Fig. 1 should also be applied for experiments in Fig. 5F.

To answer Reviewer #1's comment, we performed several additional experiments with the *Moraxella catarrhalis* strain in primary human macrophages treated with siRNA targeting ARL5b or Luciferase as a control. While we were able to observe an impairment in bacterial clearance in the HRV16-treated cells, the effect of the siRNA treatment was too variable to be conclusive, in particular in the context of these bacteria for which the intracellular survival was difficult to monitor.

Reviewer #2:

Jubrail and colleagues present interesting data in support of HRV-mediated suppression of bacterial killing my human macrophages. The experiments are appropriate, the data well presented and conclusions justified. I feel that this is of general interest to the readership of EMBO Reports and to the scientific community in general. I believe the paper can be published with only a few minor revisions:

We thank Reviewer #2 for their kind comments on our manuscript as well as the acknowledgement of the quality of the work.

1. I think the interpretation of the H2DCFDA data may be erroneous. Once the H2DCFDA is oxidized the signal cannot be reversed (ie. reduced again) in the phagosome. So the reduced signal that is observed with time is likely related to the quenching of fluorescein in the acidic compartment as opposed to ROS kinetics. Nevertheless, this is minor and does not change the major interpretation - it is still safe to say ROS is reduced.

We thank Reviewer #2 for this comment and pointing out this property of H2DCFDA. We have modified the text accordingly, which now reads as follows: "The oxidative burst was detected as soon as 10 min in control conditions with a peak at 30 min and a decline of signal by 120 min, probably due to the quenching of the fluorescein in the acidic compartment (Figure 1H)."

2. What is the significance of measuring surface CD63 by flow cytometry? Is surface CD63 relevant to phagosomal CD63?

We thank Reviewer#2 for this question. CD63 is known to traffic to the plasma membrane before being addressed to late endosomes. Therefore, analyzing surface CD63 is a way to assess the general trafficking of this protein. An increased surface level of CD63 suggests a retention of the protein at the plasma membrane and thus indicates a problem in the traffic of CD63, either accelerated secretion and/or impaired endocytosis. We have modified the text accordingly for sake of clarity.

3. The individually cropped bands of Figure 4F are a little concerning. Please provide a supplementary figure with the uncrossed gels.

We thank Reviewer #2 for this comment also raised by Reviewer #1. See response under question #9, Reviewer #1.

4. All in all, a nice paper.

We thank Reviewer #2 for this kind comment.

Reviewer #3:

In this article Jubrail et al address the problem of how human rhinoviral infections enhance susceptibility to bacterial super-infections. The question is relevant from the perspective of rhinoviral biology where many questions still remain open. It also has clinical pertinence since rhinoviral and associated bacterial infections are important precursor for asthma and COPD exacerbations in various age groups.

The authors demonstrate defective bactericidal activity of macrophages upon HRV16 exposure, and identify perturbations in the phagosome maturation pathway. With RNA-seq, they further identify gene clusters with altered expression profiles in infected macrophages and characterise the role of one of the genes identified, ARL5b in greater detail. The data suggests dual function for ARL5b in HRV infection - upregulation upon viral infection perturbs phagosome maturation in macrophages, whereas restricts virion secretion in epithelial cells.

The experimental design and execution are commendable, yielding clear results. However, I fear the strength of some of the conclusions drawn exceeds the available evidence, necessitating additional experiments. Please see my detailed comments below.

Figure 1B: The figure indicates that all infection conditions exhibited a nearly identical bacterial load at the initial time point depicted. This finding is perplexing considering the authors' previous demonstration of significant uptake defects during HRV infection, which varied for different bacterial species. In the methods section, it is mentioned that the MOI was adjusted to be four times higher for the viral-infected cases in order to achieve the same initial bacterial load (an explanation that could be included in the results section to prevent reader confusion). However, it remains unclear how this adjustment resulted in identical initial CFUs for all bacterial cases. Previous reports, for example, have shown a difference of nearly two orders of magnitude in initial CFUs between the control and MC, and nearly an order of magnitude between NTHi and MC.

We thank Reviewer #3 for this comment and the question also raised by Reviewer#1. Please see response under Question #1, Reviewer#1.

The hydrolytic activity shown in figure 1G: Taking it longer beyond the peak would help to see if it's a delayed rise of activity or reduction in peak activity. This has bearing on questions related to figure 2 mentioned below.

We agree with the Reviewer that analysing longer time points would help decipher if the differences observed between the HRV16 and MI conditions are due to a delay in the induction of the hydrolytic activity or to a reduction. To answer this question, we have performed several additional experiments and have analyzed the hydrolytic activity in phagosomes up to 6h post beads phagocytosis (new Figure 1G). These results showed that after 120 min post exposure to the beads, the fluorescence decreased in all conditions. Interestingly, we observed that HRV16-exposed macrophages showed a lower hydrolytic activity as compared with the MI treated cells, and not a delayed response, since the peak of activity was monitored at 120 min in all conditions.

Figure 2 A and C show images taken at 60 min, by which time the enrichment of the markers on phagosomes is lost in controls. Showing an earlier time point when the marker enrichment is visible in controls will help readers appreciate if the phagosomes and enrichment in controls qualitatively appear similar, to the later time point in HRV treated (the data suggests that possibility).

As suggested by Reviewer #1 and #3, we have added representative images for all the time points quantified for the different staining (new Figures 2A, 2C, 2E, 2G, and 2I).

The data presented in Figure 2 can also be interpreted as indicating a slower maturation of the phagocytic process rather than a complete blockage. In the HRV condition, EEA1 reaches its peak at 15 minutes and tends to decrease by the 30-minute mark, while CD63 appears to increase towards control levels by the 30-minute time point. To gain further insights and distinguish between potential issues with slow kinetics or aberrant marker association levels, it would be beneficial to extend the measurements to additional time points. Observing how these markers resolve towards control levels over time, will lead to better understanding of whether the observed phenomena represent slow kinetics and/or disrupted marker associations (phagosome maturation). With the present data, it is challenging to definitively conclude that phagosome maturation is completely arrested.

We thank Reviewer #3 for this important comment. To answer, we first performed additional staining at a later time point (i.e. 120 min post phagocytosis). At 120 min, the percentage of EEA1+ phagosomes remained stable in HRV16-treated cells, whereas it was decreased at 60 min and then increased at 120 min in MI conditions (Figures 2A-B). Therefore, the association of EEA1 to the phagosomes in HRV16-treated macrophages was less dynamic than in the control cells. Our results also showed that the association of CD63 to phagosomes observed in HRV16-treated macrophages, also remained through time and was still apparent at 120 min post phagocytosis initiation (Figures 2E-F). These results suggest that maturation is arrested after CD63 recruitment. This was reinforced by the analysis of LAMP1 recruitment to phagosomes that remained lower at 120 min (although not significantly) in the HRV16 condition compared to the control condition (Figures 2I-J). Altogether, these results suggest that the differences observed are due to an arrest in phagosomes maturation after CD63 recruitment. We have modified the text accordingly.

Furthermore, drawing such a definitive conclusion based solely on the three markers utilized seems somewhat overstated. Including the examination of additional markers that play a crucial role in the dynamics of phagocytosis, such as certain Rab-GTPases known to be essential in phagosome maturation, would contribute to a more accurate delineation of the process and any potential aberrations.

We thank the Reviewer for this suggestion, also made by Reviewer #1. Please see response under comment #4, Reviewer #1.

A related question pertains to phagocytosis itself. Do HRV-treated cells exhibit defects in red blood cell (RBC) phagocytosis? Are there differences in phagocytosis indices compared to controls? If such differences exist, it raises the question of whether HRV-treated cells initiate the maturation process with significantly divergent phagocytosis loads. Does it have any bearing in say, apparent slower maturation?

We thank the Reviewer for this question. We have indeed previously shown that bacterial uptake and various types of receptor-mediated phagocytosis is impaired in HRV16-treated macrophages (Figures 1 and 2, Jubrail *et al.* 2020). To assess if differences in the phagocytosis load could account for the differences observed in phagosome maturation, we performed additional experiments in which HRV16-treated macrophages were exposed to two times the amount of RBCs (condition named "HRV16 (x2 RBC)"), as compared with the control cells. Our results showed that macrophages with an increased phagocytosis load still presented increased EEA1 recruitment to phagosomes (appendix Figure 1 below). However, the phagocytosis efficiency was not increased when the cells were incubated with more RBCs, probably because the receptor-mediated phagocytosis was already at a maximum. It is therefore difficult to conclude on this point.

Of note, this question was also taken into consideration in the bacterial clearance assays presented in Figure 1, where there was indeed restoration of the bacterial uptake with more bacteria, allowing to compare the clearance activity of the cells. Please see also Response #1 under Reviewer #1.

Appendix Figure 1: Recruitment of EEA1 on phagosomes in HRV16-treated macrophages incubated with two times more RBCs. hMDMs were challenged with HRV16 or controls and then exposed to either 10 IgG-RBCs or 20 RBCs per cell for 60min, n = 3 donors, 30 random cells/coverslip. *P < 0.05 one-way ANOVA with Bonferonni's post-test vs. MI. Error bars represent standard error of the mean (SEM).

The data in figure 3 suggests more than what has been concluded. It is not merely expression and localisation of markers that has been perturbed. Especially with EEA1, it can be appreciated that the vesicles in HRV challenged cells have different characteristics. They are more heterogeneous in size, and a significant portion of the vesicles are much larger than the control vesicles with dramatically higher fluorescence. This is corroborated by the quantifications, which show that there is higher amount of EEA1 without any increase in total number of vesicles. The late endosomal marker also, less strikingly though, suggests alterations to the endosomal compartment. The late endosomes are surface associated, and the vesicle sizes, look potentially quite different.

We thank Reviewer #3 for pointing out this important point. Indeed, more heterogeneous and more intensely stained EEA1 vesicles are observed in HRV16-treated cells. It has to be noted, however, that the endosomes were not bigger, as observed at the level of electron microscopy (see response below). We have added this description in the text to better describe our results and modified the text accordingly.

While a detailed analysis might be beyond the scope of this paper, a more detailed examination of the endomembrane system using light or electron microscopy would be very beneficial. It would aid in determining whether the cell's defects are solely limited to phagosome maturation or if they indicate a broader disruption in endomembrane system organization, which in turn affects phagocytosis.

Such an analysis would also be relevant to the overall narrative as ARL5B, a Golgi-associated GTP-ase, potentially plays a role in both anterograde and retrograde traffic, thereby impacting endomembrane homeostasis. This becomes particularly intriguing considering previous research has shown that disruption of the endomembrane system and traffic can occur as a result of picornavirus infections, including HRV. Elucidating whether these disruptions contribute to bactericidal defects and subsequently increase susceptibility to bacterial infections would greatly enhance the value of this article.

We thank the reviewer for this suggestion and agree that electron microscopy allows a more detailed examination of the intracellular compartments. We performed an electron microscopy analysis on MI- and HRV16-treated human macrophages. The overall morphology of endosomal compartments, and especially of early endosomes, were not altered in HRV16-treated cells as compared to control macrophages. What was more striking are changes in the ultrastructure of the Golgi apparatus: while there are multiple Golgi compartments in primary human macrophages, their morphology was as expected, with visible stacks and vesicles. In the HRV16-treated macrophages, the Golgi apparatus/TGN were less compact, with less well-structured cisternae and associated with numerous tubulo-vesicular structures (see New Figure 3J).

In addition, we performed immunofluorescence staining of the HRV16-treated macrophages and observed an increase in the total GM130 staining associated with a decrease in the TGN46 staining, as compared with control cells (see New Figures 3K-L). Together, these results highlight that the virus induces modifications in the intracellular trafficking, probably both at the level of anterograde and retrograde trafficking, resulting in an impaired biogenesis of phagolysosomes. The text has been modified accordingly.

On a similar note, it will be good to know if the significant clusters identified by RNA-seq suggest any perturbation of the membrane trafficking pathways.

We thank the Reviewer for this comment, which was also addressed by Reviewer #1. Please see response #7 under Reviewer#1.

Does ARLB5 knockdown have any effect on HRV exposed macrophage phagocytosis? Since the ARLB5 overexpression system is already made by the authors, its effect on macrophage behavior will be a good control to see.

We agree with Reviewer #3 that overexpression of ARL5b in human macrophages would be interesting to monitor if this is sufficient to lead to similar phenotypes as exposure to the virus. We have tried to set up this experiment. Unfortunately, we were not able to obtain transient expression of the protein in macrophages without altering their viability and were therefore not able to perform these experiments.

Nevertheless, we have assessed the effect of ARL5b knockdown on phagosome maturation monitored via the recruitment of the LAMP1 marker (new Figure 5G). We observed that the recruitment of LAMP1 was restored on phagosomes in macrophages depleted for ARL5b and treated with HRV16, as compared with cells treated with control siRNA in which the virus induces a defective recruitment of LAMP1. These experiments complement the bacterial clearance assay presented in Figure 5F and extends the role of ARL5b. The text was modified accordingly.

Figure 6 - while higher cell death with reinfection is an interesting observation, one would like to see the viral titres compared by a plaque assay in HeLaOH or similar to make a conclusion that "Therefore, depletion of ARL5b in HeLa cells increases HRV16 virions secretion and viral propagation."

We agree with Reviewer #3 that a proper plaque assay would give clearer results on the effect of ARL5b depletion or overexpression on viral secretion in HeLa Ohio. Therefore, we performed a new set of experiments to generate supernatants of HeLa Ohio depleted for or overexpressing ARL5b. These supernatants were then used to assess the TCID50 in the supernatants. The TCID50 obtained were coherent with our previous results and showed that overexpressing a WT ARL5b led to a decrease of the TCID50, whereas the overexpression of an inactive form of ARL5b had no effect on the TCID50 (Figure 6D). On the other hand, depletion of ARL5b led to an increase of the TCID50 (Figure 6G). Therefore, our new experiments confirmed that depletion of ARL5b in HeLa Ohio increases HRV16 virions secretion and viral propagation.

Sentence 220-222 is written in a confusing manner with an active role ascribed to siLuciferase - it simply needs to say that control siRNA did not alter the behavior. Same issue with 225-227.

Both sentences have been modified accordingly.

In general figure explanations are somewhat inadequate. The legend and the text together still leave the figures not fully explained. As an example, the arrows in Fig 2A are not explained.

We apologize for overlooking the lack of explanation of the arrows as well as the inadequacy of the figure legend. We have revised the figure legends to increase their readability. The arrows have been removed in the modifications of the figures.

There are inconsistencies in language and usage of abbreviations that need to be taken care of.

We thank Reviewer #3 for pointing this out. We have carefully revised the test accordingly.

Dear Florence,

Thank you for the submission of your revised manuscript to EMBO reports. We asked referee #2 and #3 to assess the revised version and both referees are very positive about it and support publication.

Browsing through the manuscript myself, I noticed a few editorial things that we need before we can proceed with the official acceptance of your study:

- Please add the following funding info in the online submission system: ANR-10-INBS-04; N{degree sign}ANR-10-IDEX-0001-02 PSL. The information in the system and in the paper must match.
- Please reduce the number of keywords to 5.
- Data and code availability should be renamed to Data availability and placed after the Materials and Methods section. Lead contact is not normally part of our methods section but in this case it makes sense and may stay.
- Please ensure that the image data on Biostudies (<https://www.ebi.ac.uk/biostudies/bioimages/studies/S-BIAD931>) are available upon online publication of your manuscript.
- Please update the 'Conflict of interest' paragraph to our new 'Disclosure and competing interests statement'. For more information see <https://www.embopress.org/page/journal/14693178/authorguide#conflictsofinterest>
- Please remove the Author Contributions from the manuscript file and make sure that the author contributions in our online submission system are correct and up-to-date. The information you specified in the system will be automatically retrieved and typeset into the article. You can enter additional information in the free text box provided, if you wish.
- Tables EV1-EV3 should be renamed to Dataset EV1-EV3 (uploaded correctly, just not labeled) with the corresponding callouts updated in the text. The legends should be removed from the manuscript file, and uploaded as a separate sheet of each Excel file.
- You specify that the primers are listed in Table EV1 in the Author Checklist but such a table does not exist. It seems they are listed in the Methods section? Please clarify/update.
- Fig EV2A seems to have a low resolution. It appears blurred.
- Please upload the supplied Source data as one folder per one figure (not all figures together in one zip file).
- Please write the abstract in present tense.
- Our production/data editors have asked you to clarify several points in the figure legends (see below). Please incorporate these changes in the manuscript and return the revised file with tracked changes with your final manuscript submission:
 - a) Please note that a separate 'Data Information' section is required in the legends of all the figures." [Data information lists all information on e.g. statistics or scale bars that relates to all or several panels of the figure. Please specify subpanels if necessary.]
 - b) Please indicate the statistical test used for data analysis in the legend of figure EV2a.
- Finally, EMBO Reports papers are accompanied online by A) a short (1-2 sentences) summary of the findings and their significance, B) 2-3 bullet points highlighting key results and C) a synopsis image that is 550x300-600 pixels large (width x height) in PNG for JPG format. You can either show a model or key data in the synopsis image. Please note that the size is rather small and that text needs to be readable at the final size. Please send us this information along with the revised manuscript.
- On a different note, I would like to alert you that EMBO Press offers a new format for a video-synopsis of work published with us, which essentially is a short, author-generated film explaining the core findings in hand drawings, and, as we believe, can be very useful to increase visibility of the work. This has proven to offer a nice opportunity for exposure i.p. for the first author(s) of the study. Please see the following link for representative examples and their integration into the article web page:
https://www.embopress.org/video_synopses
<https://www.embopress.org/doi/full/10.15252/emj.2019103932>

With kind regards,

Martina

Referee #2:

I have reviewed the point-by-point rebuttal and I am very happy with the author's response to my concerns as well as the concerns of the other reviewers. I feel that this manuscript is ready and suitable for publication in EMBO Reports.

Referee #3:

Faure-Dupuy et. al., have done extensive experiments to address the points raised by the reviewers.

All the points have been satisfactorily answered and I have no remaining concerns.

The authors have addressed all minor editorial requests.

Dr. Florence Niedergang
Institut Cochin
Infection, Immunity and Inflammation
22, rue Mechain
Paris 75014
France

Dear Florence,

I am very pleased to accept your manuscript for publication in the next available issue of EMBO reports. Thank you for your contribution to our journal.

Best wishes,

Martina
